# Characterization of carbonaceous aerosols in Singapore: insight from black carbon fragments and trace metal ions detected by a soot-particle aerosol mass spectrometer

Laura-Hélèna Rivellini[1], Max Gerrit Adam[2#], Nethmi Kasthuriarachchi[2], Alex King Yin Lee[1,2]

[1] NUS Environmental Research Institute, National University of Singapore, 117411, Singapore
[2] Department of Civil and Environmental Engineering, National University of Singapore, 117576, Singapore
[#] Now at Department of Civil and Environmental Engineering, National University of Singapore, 117576, Singapore

*Correspondence to:* Alex K. Y. Lee (ceelkya@nus.edu.sg)

**Abstract.** Understanding sources and atmospheric processes that can influence the physio-chemical properties of carbonaceous aerosols are essential to evaluate their impacts on air quality and climate. However, resolving the sources, emission characteristics and aging processes of carbonaceous aerosols in complex urban environments remain challenging. In this work, a soot-particle aerosol mass spectrometer (SP-AMS) was deployed to characterize organic aerosols (OA), refractory BC (rBC) and trace metals in Singapore, a highly urbanized city with multiple local and regional air pollution sources in the tropical region. rBC ($C_1^+$–$C_9^+$) fragments and trace metals ions ($K^+$, $Na^+$, $Ni^+$, $V^+$ and $Rb^+$) were integrated into our positive matrix factorization of OA. Two types of fossil fuel combustion-related OA with different degree of oxygenation were identified. This work provides evidence that over 90% of rBC was originated from local combustion sources with a major part related to traffic and ~30% of them associated with fresh secondary organic aerosol (SOA) produced under the influences of shipping and industrial emissions activities (e.g., refineries and petrochemical plants) during daytime. The results also show that ~43% of the total rBC was emitted from local traffic, and the rest of rBC fraction due to multiple sources, including vehicular, shipping and industrial emissions, being not fully resolved. There was only a weak association between the cooking-related OA component and rBC. Although there was no observable biomass burning episode during the sampling period, $K^+$ and $Rb^+$ were mainly associated with the more-oxidized oxygenated OA component, indicating potential contributions of regional biomass burning and/or coal combustion emissions to this aged OA component. Furthermore, the aerosol pollutants transported from the industrial area and shipping ports gave higher $C_1^+/C_3^+$ and $V^+/Ni^+$ ratios than those associated with traffic. The observed association between $Na^+$ and rBC suggests that the contribution of anthropogenic emissions to total particulate sodium should not be ignored in coastal urban environments. Overall, this work demonstrates that rBC fragments and trace metal ions can improve our understanding on the sources, emissions characteristics and aging history of carbonaceous aerosol (OA and rBC) in this type of complex urban environments.

## 1. Introduction

Rapid economic growth of Southeast Asia have resulted in frequent air pollution episodes in the region, primarily due to emissions from different types of fuel and biomass combustions (UN Environment, 2014). Besides typical urban emissions (e.g., local traffic), Singapore is a highly urbanized and densely populated city located at the south end of the Malaysian Peninsula that can be seasonally impacted by air pollutants, especially fine particulate matter, caused by agricultural burning and wildfire events from the neighboring countries (Atwood et al., 2013; Salinas et al., 2013). Singapore is the major maritime hub in Southeast Asia. According to the World Shipping Council, Singapore has the world's second-busiest port in terms of shipping tonnage (World Shipping Council, 2019). Deep-sea cargo ships typically burn heavy residual oil and hence the impacts of ship emissions on both local and regional particulate pollution can be substantial (Saputra et al., 2013; Velasco and Roth, 2012). Furthermore, one of the world's largest oil refinery and manufacturing complexes is located in the industrial zone of Singapore (Chou et al., 2019; Diez et al., 2019) but its potential influences on local and regional air quality remains poorly understood.

Most of the air quality studies focused on particulate pollution in Singapore have been relying on off-line chemical speciation of aerosol filter samples (Balasubramanian and Qian, 2004; Atwood et al., 2013; Yang et al., 2013; Engling et al., 2014). Although this approach can provide valuable information for characterizing sources and transports of atmospheric aerosols, higher time-resolution measurements are essential for evaluating potential impacts of short-term pollution events on air quality and tracking rapid variations of aerosol compositions due to emissions and atmospheric dynamics. In particular, a combination of real-time aerosol mass spectrometry technique and positive matrix factorization (PMF) analysis has been widely used for characterizing chemical compositions and potential sources of atmospheric submicron organic aerosols (OA) worldwide (Zhang et al., 2007b; Jimenez et al., 2009). Recently, Budisulistiorini et al. (2018) observed a strong haze event that lasted for only a few hours in Singapore due to the regional transports of air mass influenced by the wildfires in Kalimantan and Sumatra, Indonesia, in 2015 using an online aerosol chemical speciation monitor (ACSM). Two OA components caused by peat and biomass burning events, accounting for ~30% of total OA mass, could be identified based on their PMF analysis.

Combustion processes are known sources of urban carbonaceous aerosol, consisting of OA and black carbon (BC). The diversity of anthropogenic activities in Singapore can lead to complex mixtures of OA and BC with their sources being difficult to be interpreted based on the PMF analysis of OA fragments measured by on-line aerosol mass spectrometry alone. For example, although Budisulistiorini et al. (2018) could identify a primary OA (POA) component originated from local fossil fuel combustion and a secondary OA (SOA) component produced through a few possible formation mechanisms in Singapore during the haze period in 2015, only limited information on their emission characteristics and formation/aging processes could be provided. Taking the advantage of soot-particle aerosol mass spectrometer (SP-AMS) that can detect refractory BC (rBC, an operational defined term) and trace metals in addition to the non-refractory particulate matter (NR-PM, including OA, sulfate, nitrate, ammonium and chloride) (Carbone et al., 2015; Onasch et al., 2012), the primary goal of this work is to improve our understanding on the emissions characteristics and aging processes of carbonaceous aerosols in Singapore by performing three versions of PMF analysis with different model inputs obtained from our SP-AMS measurements: 1) OA fragments, 2) OA and rBC fragments ($C_n^+$), and 3) OA and rBC fragments and trace metal

ions. This is the first time to deploy a SP-AMS in Singapore and Southeast Asia for this type of field investigation. The results provide insights into how OA components are co-emitted and interact with BC and metal species in the atmosphere.

## 2. Methodology

### 2.1 Sampling site and co-located measurements

A SP-AMS was deployed to measure the chemical composition of atmospheric submicron aerosols (see details in Section 2.2). Other co-located instruments for $PM_{2.5}$ characterizations include an aethalometer (AE33, Magee Scientific), a Monitor for Aerosols and Gases (MARGA, Metrohm) and a semi-continuous organic and elemental carbon (OC/EC) analyser (Sunset Laboratory). Gas-phase species, including nitrogen oxides ($NO_X$, Horiba APNA-370), carbon monoxide (CO, Horiba APMA-370), and ozone ($O_3$, 2B Technologies Inc., Model 202), were also measured. The measurements were carried out from 14 May to 9 June 2017 at the E2 building (1°18'N, 103°46', 67 m above sea level) within the campus of the National University of Singapore. The measurement site is located at the southern part of Singapore, which is ~1 km away from the large-scale shipping port facilities (Figure S1a). The air quality at the site can be influenced by nearby traffic and emissions from an industrial area located at the southwest of the measurement site (Table S1).

Co-located meteorological measurements of wind speed and direction (RM Young model wind sentry set 03001), relative humidity (RH) and temperature (Vaisala model CS500), solar radiation (LI-COR model LI-200X), and rainfall (hydrological services CS700) were conducted throughout the entire sampling period. In general, the month of May falls into the inter-monsoonal period (i.e., March-May) characterized by low wind speed, whereas June to September are classified as the southwest monsoon period characterized by low rainfall levels compared to the rest of the year (NEA, 2018). During this field study, the shift from low intensity and variable winds toward stronger wind blowing from southwest was observed during the second half of May. Furthermore, the beginning of June clearly appeared under low level winds from southwest (Figure 1 and S1b). These observations suggest that the sampling period covered the transition period from late inter-monsoon to the early southwest monsoon season.

### 2.2. Data collection

#### 2.2.1 Soot-particle aerosol mass spectrometer (SP-AMS)

A Teflon-coated cyclone (URG, model 2000-30ED) with a cut-off diameter of 2.5 μm was installed in the sampling inlet of the SP-AMS. The detail description of SP-AMS has been reported by Onasch et al. (2012), and thus only a brief description of its working principle is given in this section. SP-AMS is based on the design of high-resolution time-of-flight aerosol mass spectrometer (HR-ToF-MS, DeCarlo et al., 2006). HR-ToF-AMS can quantify NR-PM by flash vaporization of aerosol particles at 600°C on a resistively heated tungsten surface. SP-AMS has an additional Nd-YAG intra-cavity infrared (IR) laser module at the wavelength of 1064nm. rBC is strongly absorbing in this wavelength, hence heating up to ~ 4000°C for rBC vaporization (Onasch et al., 2012). The term rBC is an operationally defined term that applies to BC particles detected by SP-AMS. The resulting

gas-phase molecules are ionized by electron impact (EI) ionization method at 70 eV and analyzed by a ToF-MS. Non-refractory and refractory species that are internally mixed with soot particles, such as organics, inorganics, or trace elements, can be detected through laser vaporizer measurements (Carbone et al., 2015; Corbin et al., 2018; Onasch et al., 2012). In the present study, the SP-AMS was operated at 1 min alternating intervals between IR laser-on (i.e., dual vaporizers) and laser-off mode (i.e., tungsten vaporizer only), with a mass spectral resolving power of approximately 2000 at $m/z$ 28. The vacuum aerodynamic diameter ($d_{va}$) of ambient particles is determined by the measured particle time-of-flight (PToF) from the chopper wheel to the tungsten vaporizer (Jayne et al., 2000).

### 2.2.2 Calibrations

Calibrations of the SP-AMS for quantifying NR-PM in the laser-off mode were performed by generating dried monodispersed (300nm) ammonium nitrate ($NH_4NO_3$) particles. The $NH_4NO_3$ solution was atomized by a constant output atomizer (TSI, model 3076). The aqueous droplets were subsequently dried by passing through a diffusion dryer and were size-selected using a differential mobility analyzer (DMA, TSI Inc., Model 3081). The ionization efficiency ($IE_{NO3}$) and the mass-based ionization efficiency ($mIE_{NO3}$) were determined based on three sets of calibration data throughout the sampling period. Similarly, dried mono-dispersed (300nm) black carbon particles generated by atomizing the standard Regal Black (Regal 400R Pigment Black, Cabot Corp., recommended by Onasch et al., 2012) were used to determine the mass-based ionization efficiency of rBC ($mIE_{rBC}$) and its ionization efficiency relative to nitrate ($RIE_{rBC} = mIE_{rBC}/mIE_{NO3}$) for the quantification of rBC in the laser-on mode.

The calibration and field data were processed using the IGOR-based AMS data analysis software SQUIRREL v. 1.16I for unit mass resolution (UMR) data and PIKA, v. 1.57I for high resolution peak fitting (Sueper, 2015) with the corrected air fragment column of the standard fragmentation table based on particle-free ambient air data (Allan et al., 2004; DeCarlo et al., 2006). The sum of carbon ion clusters $C_1^+$-$C_9^+$ measured in the laser-on mode was used for quantifying rBC mass in both calibration and ambient data. The average $C_1^+/C_3^+$ ratio (0.65) obtained from Regal Black was used to correct the non-refractory organic interference in $C_1^+$ for quantification of total rBC mass loadings. The campaign averages of $RIE_{rBC}$ and $RIE_{NH4}$ were 0.15 ($\pm$0.04) and 4.24 ($\pm$0.04), respectively. The default RIE values of nitrate (1.1), sulfate (1.2) and organics (1.4) were used for respective mass concentration quantification (Jimenez, 2003). NR-PM were quantified based on the laser-off mode measurements and their CE were determined using the composition-dependent approach (CDCE, Figure S2a) (Middlebrook et al., 2012). Note that the sample stream was dried to less than 30% RH by a Nafion dryer (PD-200T-12 MPS, Perma Pure) throughout the sampling period in order to reduce CE uncertainties due to particle bouncing on the surface of tungsten vaporizer. The SP-AMS measurements were compared with the sulfate and OM concentrations measured by the MARGA and the OC/EC analyser, respectively, showing strong temporal (r = 0.77 and 0.93) and quantitative (slopes = 0.81 and 0.88) agreements between these measurements (Figure S3a and b). A collection efficiency (CE) of 0.6 was used for rBC quantification due to incomplete overlap between the laser vaporizer and the particle beam (Willis et al., 2014). The corrected rBC concentrations were also comparable to those measured by the aethalometer (r = 0.96, slope = 0.83) and the EC measured by the OC/EC analyser (r = 0.84, slope = 1.1)

(Figure S3c and d). Potassium ($K^+$), sodium ($Na^+$), vanadium ($V^+$), nickel ($Ni^+$) and rubidium ($Rb^+$) ions were detected (Table S2). The signals of these trace metal ions were not calibrated and thus their raw signals were used for investigating their temporal variations.

## 2.3 Data analysis

### 2.3.1 Positive matrix factorization of OA, rBC and metals

Positive matrix factorization (PMF) is widely used for identifying sources of OA measured by different versions of aerosol mass spectrometers (Ulbrich et al., 2009; Zhang et al., 2011). Each factor contains signals at *m/z* values, which have common source characteristics and chemical properties that can be used to trace the origin and processing history of that factor. The PMF evaluation tool (PET, v3.00D) (Ulbrich et al., 2009) was used to identify the potential sources and characteristics of OA based on the organic fragments measured in the laser-off mode. In the process of applying PMF, the ions with a signal-to-noise ratio (SNR) of $0.2 < SNR < 2.0$ were downweighted by a factor of 2. $CO_2^+$ related ions ($O^+$, $HO^+$, $H_2O^+$, $CO^+$) were downweighted and bad ions (SNR $< 0.2$) were removed from the analysis. The results were obtained for 8 up to 10 factors with f-peak varying from -1 to 1 and increasing in step size of 0.2. Elemental ratios (O:C, H:C and N:C) of each factor were determined using the improved-ambient method as described by Canagaratna et al. (2015b). A five-factor solution was selected as final result from OA laser-off PMF analysis.

PMF analysis was further applied to determine how rBC associate with different OA factors. For this purpose, the PMF inputs were generated using 1) organic fragments measured by the laser-on mode (Figure S4a-c) and 2) both organic and $C_n^+$ fragments measured by the laser-on mode (Figure S4d-f). The $C_n^+$ in the mass spectra of each PMF factor were used to quantify the fraction of rBC associated with the identified OA components. Note that the interference of non-refractory organic signals on refractory $C_n^+$ fragments were subtracted based on the method described in Wang et al. (2018), and hence the $C_1^+$ fragment was not corrected based on the $C_3^+$ fragment, in the PMF analysis. Lastly, five metal ions ($K^+$, $Na^+$, $Ni^+$, $V^+$, $Rb^+$) were included into the PMF model (Figure S4g-i) as the majority of their signals were higher than their respective limit of detection (Table S2). The metal ion signals were corrected to nitrate equivalent mass concentrations by assuming their RIE values equal to 1 and the $K^+$ signals were downweight by a factor 2. A similar approach was applied by Carbone et al. (2019) including rBC fragments, calcium ($^{42}Ca^+$) and zinc ($^{68}Zn^+$) isotopic ions, to distinguish between lubricating oil and fuel OA types emitted at the exhaust of diesel engines. Note that the correction approach for obtaining refractory $C_n^+$ signals can only be applied when both mass spectra and time series of each OA component identified in the laser-off and laser-on PMF analysis are similar. The scatter plots (Figure S5) and correlation coefficients (r) (Table S3) highlight the consistency of the mass spectra ($0.91 < r < 0.99$), as well as time series ($0.78 < r < 0.98$), of the five OA types identified in this work. The time dependent collection efficiency (or CDCE) of rBC were determined based on the BC measured by the aethalometer (Figure S2b). Additional PMF analysis, with CDCE applied on the input matrix for both OA and refractory components, were conducted (see details in Text S1 of the supplementary information). The PMF results were compared with those obtained without CDCE applied on the input matrix as shown in Table 1 and S4.

**2.3.2 Air mass back-trajectories and wind analysis**

The origins of air masses were investigated through 5 days back trajectories generated every hour (648 back trajectories in total) at the height of 64 m using the Hybrid Single-Particle Lagrangian Integrated Trajectory (HYSPLIT) model coupled with meteorological data from the Global Data Assimilation System (GDAS, 1°). This model was developed by the National Oceanic and Atmospheric Administration (NOAA) (Draxler and Rolph, 2003). A cluster analysis was performed on all the hourly back trajectories and the three identified clusters are shown in Figure S1c. The detail of the back-trajectories clustering can be find in Baker (2010) and Borge et al. (2007).

The potential sources of pollutants were investigated by using ZeFir non-parametric wind regression (NWR) and potential source contribution function (PSCF) package developed by Petit et al., (2017) under Igor Pro. These analytical methods require high time-resolution atmospheric measurements as inputs and combine them with local wind measurements for NWR and air mass back-trajectories for PSCF. The NWR is commonly employed to identify nearby sources by coupling atmospheric species concentrations with co-located wind speed and direction (Henry et al., 2009). This method consists of weighting average concentrations with each wind speed and direction couples and using two kernel smoothing functions. In this study, the NWR graphs were generated for an angle resolution of 1° and a radial resolution of 0.1 m s$^{-1}$ and their respective smoothing parameters (or kernel parameters) have been empirically set at 5 and 2.5. For regional sources investigation, the PSCF has been favored as larger geographical scales can be studied (Polissar et al., 2001). The PSCF principle consists of redistributing high pollutant concentrations based on air mass trajectories residence time. Thus, individual species concentrations are redistributed along the trajectories into geographical emission parcels. In our study, PSCF calculation were performed considering only pollutant concentrations above their respective 75$^{th}$ percentile (graphic parameters: cell size = 0.1°, smoothing = 2).

## 3    Results and discussion

### 3.1    Overview of SP-AMS measurements

Figure 1a presents the time series of particle- and gas-phase species, PMF factors and meteorological parameters (i.e., solar radiation, temperature, RH, precipitation, wind direction and wind speed) measured during this field campaign. The mean concentration of total PM (i.e., NR-PM + rBC measured by the SP-AMS) observed in this campaign was 11.9 ($\pm$ 5.0) $\mu$g m$^{-3}$, which is about one-fifth of the average concentration of 49.8 $\mu$g m$^{-3}$ during the haze event due to the Indonesian wildfires in 2015 (Budisulistiorini et al., 2018). The main contributors to the submicronic aerosol mass loadings were OA (5.59 $\pm$ 2.66 $\mu$g m$^3$; 46.6%), followed by sulfate (SO$_4^{2-}$, 3.29 $\pm$ 1.83 $\mu$g m$^3$, 27.4%) and rBC (1.79 $\pm$ 0.37 $\mu$g m$^3$; 17.6%), whereas ammonium (NH$_4^+$, 0.85 $\pm$ 0.37 $\mu$g m$^3$; 7.1%), nitrate (NO$_3^-$, 0.11 $\pm$ 0.09 $\mu$g m$^3$; 0.9%) and chloride (Cl$^-$, 0.05 $\pm$ 0.04 $\mu$g m$^3$; 0.4%) were minor contributors (Figure 1b). The overall chemical compositions were not sensitive to the three major types of air mass back trajectories identified within the sampling period (Figure S1d).

The average RH was ~75% for temperature ranging from 26 to 32°C. The time series of wind speed and direction show a regular pattern with stronger wind speed from the southwest direction and drier condition between 13:00 and 16:00 local time (LT) (Figure S1b). This is primarily due to sea breeze phenomena that are commonly observed in Singapore during inter-monsoon period. Note that a large industrial zone is located at the southwest of the measurement site (Figure S1a). Sea breeze might carry polluted air from industry, such as oil refinery and shipping emissions, to the measurement site and the surrounding area. In particular, rBC is considered as a primary and persistent air pollutant emitted from combustion processes. While the rBC hotspot observed under low wind speed conditions was likely due to local traffic emissions, the elevated levels of rBC in the southwest sector (Figure 2) highlight the potential influences of industrial emissions of both primary and secondary pollutants on the air quality of the sampling region as discussed in Section 3.2 and 3.3. The detail of rBC sources and characteristics are discussed in Section 3.4 and 3.5.

**3.2 Acidic sulfate and organo-sulfur formation during sea breeze period**

The diurnal cycle of $SO_4^{2-}$ shows a sharp increase starting from ~10:00 LT. reaching a maximum at ~15:00 LT (Figure 1c), which can be explained by the active photochemistry during daytime. The average value of 0.71 for the measured to predicted $NH_4^+$ ratio ($NH_4^+_{,meas}/NH_4^+_{,pred}$) (Zhang et al., 2007a) and the absence of diurnal pattern for $NH_4^+$ (Figure 1c) implies that gaseous ammonia were insufficient to neutralize acidic sulfate plumes within a timescale of a few hours (Attwood et al., 2014; Guo et al., 2015; Kim et al., 2015). Note that $NO_3^-$ and $Cl^-$ mass concentrations were too low to have substantial influences on the $NH_4^+$ levels. The diurnal cycle and NWR plot of $NH_4^+_{,meas}/NH_4^+_{,pred}$ ratios (Figure S6a and b) illustrate that the sulfate plumes observed during the sea breeze from southwest were more acidic than the background sulfate aerosol (Figure S6c). This particle acidity dependence seems to coincide with diurnal variations of $SO_4^{2-}$ size distribution. Throughout the day, relatively large sulfate particles are observed with their vacuum aerodynamic diameter ($d_{va}$) peaked at ~400 nm, whereas those encountered during the sea breeze present a broader mode with $d_{va}$ ranging between 200 and 400 nm (Figure S6a and d). This suggests that a large fraction of acidic $SO_4^{2-}$ particles observed during the sea breeze period were freshly formed in the atmosphere.

It is important to note that the period with elevated concentrations of sulfate and sulfur-containing fragments were synchronized with the sea breeze from the southwest, from which refineries, petrochemical industries and shipping emissions could be carried to our sampling site (Figure 2 and S6e). Therefore, the enhancements of sulfate and organo-sulfur could be due to oxidation of biogenic dimethyl sulfide (DMS) emitted from the ocean which can produce methylsulfonic acid (MSA) and $SO_4^{2-}$ (Ge et al., 2012; Willis et al., 2016; Xu et al., 2017; Saarikoski et al., 2019) in addition to the local anthropogenic sources such as $SO_2$ emitted from shipping activities and industries (e.g., oil refinery in this study) (Saliba et al., 2010; Ripoll et al., 2015; Rivellini et al., 2017). The sulfur-containing fragments measured by the SP-AMS, including $CH_3SO^+$, $CH_2SO_2^+$ and $CH_3SO_2^+$ that could be attributed to both MSA and organo-sulfur compounds (Farmer et al., 2010). In particular, $CH_3SO_2^+$ fragment showed a moderate correlation with both $SO_4^{2-}$ and less-oxidized oxygenated OA (LO-OOA, see PMF results for OA in Section 3.3) (r = 0.66-0.73) and the lowest $NH_4^+_{,meas}/NH_4^+_{,pred}$ values appeared to coincide with the highest values of $CH_2SO^+$ signals (Figure S4).

### 3.3 Identification of major OA sources

Five distinct OA components were identified based on the PMF analysis of organic fragments measured by the laser-off mode of SP-AMS measurement (i.e., a standard HR-ToF-AMS measurement). The OA components include hydrocarbon-like OA (HOA), oxygenated-HOA (O-HOA), cooking-related OA (COA), less-oxidized oxygenated OA (LO-OOA) and more-oxidized oxygenated OA (MO-OOA). The mass spectra and diurnal cycles of the five OA components are presented in Figure 3. The OOA components were the main contributors to the total OA, with MO-OOA and LO-OOA accounting for 32.1 and 10.4%, respectively. The HOA, O-HOA and COA components contributed to 19.4, 26.4 and 11.7% of the total OA mass, respectively (Figure 1b). The detailed descriptions of each OA factor, classified into 1) combustion emissions, 2) cooking emissions and 3) secondary organic aerosol, are discussed in the following sections.

### 3.3.1 Combustion emissions

HOA was the main contributor to the POA fraction and moderately correlated with the combustion tracers including rBC ($r = 0.79$), $NO_x$ ($r = 0.83$) and CO ($r = 0.81$). The mass spectrum of HOA is similar to those from engine exhausts (Crippa et al., 2013; Ng et al., 2011; Zhang et al., 2005), and is dominated by non-oxygenated fragments (i.e., $C_xH_y$) with an average H:C and O:C ratios of 2.12 and 0.07, respectively. HOA accounted for up to ~20% of total OA during the morning (~9:00 LT) which matched the enhancement of traffic volume during the rush hours. In addition, the concentrations of HOA started increasing at the beginning of evening rush hours (~17:00 LT) and reached the maximum at ~21:00 LT. HOA and combustion tracers (rBC, $NO_x$ and CO) exhibited a hotspot in the NWR plots (Figure 2, S7e and f) at relatively low wind speed conditions. The above observations suggest that HOA concentrations were largely influenced by nearby on-road engine exhausts and boundary layer contraction in the evening.

High concentrations of rBC, $NO_x$ and HOA were observed during the mid-night between May 25$^{th}$ and 28$^{th}$, indicating the presence of large combustion sources during nighttime that might impact the air quality at our sampling site occasionally (Figure 1a). This observation also reflected from the diurnal plots (Figures 3 and S8) that the mean mass concentrations of HOA, rBC, $NO_X$ and CO during the mid-night were much higher than their corresponding median values. Figure S8a shows that the highest N:C ratio was observed during the morning traffic peak hours but remained relatively low for the rest of the period. Such observation suggests that the emission characteristics of these unknown combustion sources at nighttime could be different to those associated with traffic emissions during the daytime. Although it is difficult to confirm the origin of those nighttime combustion emissions without further evidence, the observed events were coupled to low wind speed condition, suggesting that they were likely due to local emissions from the city. For example, significant emissions due to flaring from petrochemical industries, which largely depends on plant operation, during the relatively stagnant atmospheric condition could lead to elevated concentrations of the combustion-related species.

O-HOA showed distinctive time series and oxygenation level compared to HOA ($r = 0.17$, O:C = 0.07). The mass spectral profile (Figure 3) illustrate that O-HOA could be an oxygenated fraction of combustion particles which

were co-emitted with HOA and/or produced rapidly through oxidation of HOA near emission sources. However, similar to the case of rBC, the elevated concentrations of O-HOA observed from the southwest direction with strong wind speed ($\sim$18 m s$^{-1}$) suggests that O-HOA was partly related to the emissions transported from the industrial zone (Figure 2). The temporal variability of O-HOA depicts a rather singular behavior, showing a higher average concentration before May 24$^{th}$ (2.2 µg m$^{-3}$). During rest of the campaign, the principal source of O-HOA was mainly local anthropogenic activities (Figure S7a-d) and remained at rather low concentration with an average of 0.9 µg m$^{-3}$. The relatively constant diurnal cycle of O-HOA compared to that of HOA further underlines multiple sources of this OA component.

### 3.3.2 Cooking emissions

COA is another major POA component identified in this study. COA concentrations remained low in the morning and exhibited peaks around lunch (13:00 LT) and dinner time (21:00 LT) (Figure 3). This type of COA diurnal pattern has been commonly observed in urban studies (Allan et al., 2010; Sun et al., 2011; Fröhlich et al., 2015). Furthermore, COA mass spectrum shows a higher $m/z$ 55/57 ratio than HOA as reported in previous studies (Allan et al., 2010; Mohr et al., 2012) and is strongly correlated with the COA mass spectrum reported in France (Crippa et al., 2013) and in the megacities of China (Hu et al., 2013; Elser et al., 2016). An organic fragment tracer, $C_6H_{10}O^+$, tracked the time profile of COA ($r^2 = 0.90$), further confirming the existence of COA components (Zhang et al., 2011). A local COA hotspot shown in the NWR plots (Figure 2) suggests that the campus canteens and residential building nearby are likely the major contributors to the observed COA mass loadings. Note that COA can contribute up to 50% of total OA in other urban locations (e.g., downtown area with substantial emissions from restaurants) depending on the site characteristics (Allan et al., 2010; Huang et al., 2010; Sun et al., 2012; Crippa et al., 2013) and cooking styles. Similar to other combustion-related emissions, high concentrations of COA component were also observed during the mid-night between May 25th and 28$^{th}$ (Figure 1). This observation implies that combustion-related emissions might contain some COA-like particles, contributing to the reported mass concentrations of COA in general. After removing the data from May 25th and 28$^{th}$, the diurnal profile shows that lower mass concentrations of COA were observed overnight (Figure S8d).

### 3.3.3 Secondary organic aerosol

The mass spectra of LO-OOA was dominated by the oxygenated fragments of $C_2H_3O^+$ and $CO_2^+$ whereas MO-OOA was mainly characterized by $CO_2^+$ fragments (Figure 3). The different degree of oxygenation between LO-OOA and MO-OOA and their secondary nature are shown in the Van-Krevelen diagram (Figure S9) (Heald et al., 2010; Ng et al., 2011). MO-OOA (O:C = 0.94, N:C = 0.01) likely represented a more aged fraction of SOA. The MO-OOA concentrations were insensitive to the local wind pattern (i.e., more homogeneous distribution in the NWR plot, Figure 2) and increased gradually in the afternoon, possibly due to the combined effects of photochemistry and mixing of air masses transported from the regions without significant local industrial influences. Similar to many other studies, sources identification of MO-OOA component is not straightforward as their mass spectral features can be obtained by oxidation of different types of POA and SOA components

(Donahue et al., 2009; Jimenez et al., 2009; Zhang et al., 2007b). Additional information is essential to further interpret their potential sources and/or formation mechanisms (see discussion in Section 3.5).

In contrast, the moderate correlation observed between LO-OOA and $SO_4^{2-}$ (r = 0.73) and the similarity between their diurnal patterns illustrate that LO-OOA were freshly formed SOA materials (O:C = 0.41). The latter might be formed via local photo-oxidation of POA and/or SOA precursors under the influences of anthropogenic emissions transported form the southwest direction due to sea breeze (Figure 2). Previous laboratory studies have evidenced the importance of acidic seeds and RH on SOA formation (Liggio and Li, 2006; Wong et al., 2015). The acidic nature of sulfate particles and RH encountered on site might facilitate the partitioning of volatile organic compounds (VOCs, from both biogenic and anthropogenic sources) into particle-phase, leading to the production of SOA and perhaps organo-sulfur compounds as discussed in Section 3.2. Furthermore, Kasthuriarachchi et al., (2020) reported that the concentrations of organo-nitrate and nitrogen-containing fragments (i.e., $C_xH_yN^+$ and $C_xH_yNO_z^+$) slightly increased during daytime (e.g., Figure S8a) with LO-OOA, contributing to 30% of observed $C_xH_yNO_z^+$ fragments (Figure S10). This suggests that photo-oxidation of VOCs under high-$NO_X$ condition could be a potential pathway toward LO-OOA formation given that the $NO_X$ concentrations reached up to 160 ppb (Figure 1a, mean = 16.5 ppb) during the campaign.

## 3.4 Characterization of rBC associated with different OA sources

Integrating rBC fragments ($C_1^+$–$C_9^+$) to the PMF analysis of organic fragments measured by the laser-on mode of SP-AMS measurements yielded a five-factor solution. They show similar mass spectra and temporal variabilities compared to their corresponding OA factors identified in the PMF analysis of the laser-off datasets (Figures S4 and S5). These observations suggest that the major OA sources identified in Section 3.3 are still valid, and hence the addition of rBC fragments in the PMF analysis can provide additional information to improve our understanding on how ambient rBC might be co-emitted or interacted with different types of POA and SOA during their atmospheric dispersion. Note that Figure S4 shows higher mass loadings of PMF factors determined by the laser-on measurement than those observed in the PMF results from the laser-off measurement. This type of OA signal enhancements has been observed in previous studies (Lee et al., 2015; Willis et al., 2014) but the fundamental reason remains unclear. Therefore, this section will primarily focus on 1) quantifying the distribution of rBC fragments among different PMF factors and 2) investigating the potential application of rBC fragment ratios for identifying the origin of ambient rBC-containing particles.

Figure 4a shows that 43% of total rBC mass was associated with the HOA components in this study, indicating that a majority of rBC were emitted by local vehicular emissions. The COA factor was associated with less than 2% of total rBC mass, suggesting that they were unlikely to be co-emitted from modern kitchens and that coalescence of COA and rBC particles was insignificant near the sampling location, which is consistent with previous studies that COA and rBC were largely externally mixed in different urban environments (Lee et al., 2015, 2017; Wang et al., 2018). This observation also suggests that the potential interferences of combustion-related emissions (Section 3.3.2) to the overall temporal variation and mass concentrations of COA are unlikely significant. The two OA components that were influenced by local industries, O-HOA and LO-OOA, were

associated with 20 and 29% of the total rBC, respectively. This result highlights the fact that both O-HOA and LO-OOA components consist of mixtures of primary and secondary particles (i.e., rBC + HOA + OOA for O-HOA and rBC + OOA for LO-OOA) although they could be influenced by different types of industrial emissions and atmospheric processing. It is worth noting that LO-OOA had the second largest contribution to the total rBC, suggesting that at least a fraction of rBC from industrial emissions could act as effective condensation sinks of LO-OOA produced via the photochemistry along their dispersion. While MO-OOA had the largest contribution to total OA mass, only 6% of the total rBC mass was associated with this aged SOA component.

Previous studies have reported that physical structure and chemical composition of BC-containing particles depend on the conditions of combustion processes and the types of combustible involved (Vander Wal and Tomasek, 2004). Such relationship is possibly observed through the relative intensities of the carbon fragments ($C_1^+$-$C_3^+$) (Corbin et al., 2014). Table 1 shows that the $C_2^+/C_3^+$ ratios remained roughly constant regardless of the origin of rBC, whereas the $C_1^+/C_3^+$ ratios varied between PMF factors. The relative contributions of each factor to $C_1^+$ to $C_3^+$ fragments are presented in Figure 4b. The $C_1^+/C_3^+$ ratios of rBC associated with HOA was 0.66 (±0.07) , which was within the range of those emitted from aircraft-turbine, regal black (i.e., a BC standard for SP-AMS calibration) and particles produced by a propane diffusion flame (0.50 – 0.78) and, more importantly, close to those reported for diesel engine exhaust (Carbone et al., 2019; Corbin et al., 2014; Onasch et al., 2012). Furthermore, the size distribution of unit mass resolution data shows lower *m/z* 12-to-*m/z* 36 ratios (a proxy for $C_1^+/C_3^+$) for particles with $d_{va}$ smaller than 100 nm (Figure S11a). This suggests that rBC particles with relatively small diameter were mainly associated with fresh traffic emissions (Massoli et al., 2012). Our results illustrate that rBC transported from industrial area and shipping ports gave $C_1^+/C_3^+$ ratios closer to unity (i.e., LO-OOA = 0.79 (±0.10) and O-HOA = 1.00 (±0.11), which is similar to the previous observations for soot particles emitted from a marine engine using heavy-fuel-oil (Corbin et al., 2018), rBC-containing particles emitted from chemical and petrochemical industries (Wang et al., 2018), and rBC with high fullerene content (Canagaratna et al., 2015a; Corbin et al., 2014). The NWR and diurnal plots of $C_1^+/C_3^+$ ratio (Figure 4c and d) clearly show that rBC with higher $C_1^+/C_3^+$ ratios were transported to the site by sea breeze from the southwest direction, which are consistent with our PMF results that O-HOA and LO-OOA were influenced by industrial emissions and were associated with rBC with higher $C_1^+/C_3^+$ ratios compared to other OA components.

### 3.5  Characterization of metal ions associated with different OA sources

In addition to quantifying rBC, the laser-on mode of SP-AMS measurement can detect trace metals as rBC-containing particles can be heated to reach the temperature for vaporizing the associated trace metals (Carbone et al., 2015; Corbin et al., 2018; Onasch et al., 2012). In this work, five trace metal ions, including $K^+$, $Rb^+$, $Na^+$, $V^+$ and $Ni^+$, were integrated into the PMF analysis of organic and rBC fragments measured by the laser-on mode of SP-AMS. The addition of these trace metal ions did not result in major changes in the mass spectral features and temporal variations of the five PMF factors identified in Section 3.4 (Figures S4 and S5). Since these trace metals are relatively stable in the particle-phase, investigating how these trace metal ions are associated with different PMF factors can improve our understanding on the emission characteristics and perhaps the aging history of

specific OA components. This section will mainly focus on discussing the trace metals associated with OA and rBC that were associated with 1) local combustion emissions, and 2) aged SOA component (i.e., MO-OOA).

### 3.5.1 Combustion emissions

Based on the PMF results that includes trace metal ions, sodium ($Na^+$) was mainly associated with HOA, O-HOA and LO-OOA (Figure 5a) that could be due to different types of fossil fuel combustion emissions (e.g., local traffic, shipping, and various industrial activities) as discussed in Section 3.4. The $Na^+$ signals also correlated strongly with rBC (r = 0.80, Figure 5b) and moderately with HOA (r = 0.65). The absence of correlation between rBC and $Na^+$ in the laser-off mode measurement suggests a large degree of internal mixing of Na and rBC. The presence of sodium compounds in fuel, including remaining catalyst used for biodiesel esterification, drying agents, corrosion inhibitors and fuel additives, has been previously observed as a cause of fuel injectors fouling (Barker et al., 2013; Coordinating Research Council, 2013), and hence co-emissions of sodium with rBC and HOA from on-road engines was highly possible. Recent experiments conducted with SP-AMS measurements also reported non-negligible amounts of Na in soot particles emitted by marine, locomotive and vehicle engines (Dallmann et al., 2014; Omidvarborna et al., 2016; Saarikoski et al., 2017; Corbin et al., 2018; Carbone et al., 2019). $Na^+$ is widely used for identifying the influence of marine sources in source apportionment analysis, hence the potential contribution of sea spray aerosols to $Na^+$ signals cannot be neglected, especially for those factors (i.e., O-HOA and LO-OOA) connected to sea breeze transport. However, it is important to emphasize that $Na^+$ and $Cl^-$ exhibit rather poor temporal correlations (r < 0.30) for both laser-off and laser-on data regardless the influences of sea breeze. Biomass burning can be a possible source of $Na^+$ (Hsu et al., 2011) but no major fresh biomass burning emissions were observed in this study. The MO-OOA factor is suspected to be more influenced by aged regional biomass burning emissions (see more discussion in Section 3.5.2) but $Na^+$ was not strongly associated with this factor.

Vanadium (V) and Nickel (Ni) are usually residual trace metals in fuel and their ratio in emissions can vary with fuel types or combustion processes (Moldanová et al., 2009; Yakubov et al., 2016). This ratio can be used to trace emissions from ship or heavy oil combustion industries (Ault et al., 2010; Liu et al., 2017). In particular, Figure 5c shows that the $V^+/Ni^+$ ratio exhibited a distinctive diurnal variation, with higher values from 12:00 to 18:00 LT, which coincided with the sea breeze pattern. Two maxima are observed at 13:00 and 17:00 LT with the hourly-averaged $V^+/Ni^+$ ratio between 5 and 7 (75th percentiles > 12) (Figure 5c). Assuming similar RIE for $V^+$ and $Ni^+$ measured by the SP-AMS, our observation is consistent with other studies, reporting that the $V^+/Ni^+$ ratios ranged from 4 to 7 for ship emissions and heavy fuel combustion by engines (Agrawal et al., 2009; Corbin et al., 2018; Viana et al., 2008). In addition to a moderate correlation between LO-OOA and $V^+$ (r = 0.47), ~70% of the $V^+$ signals were associated with the LO-OOA component (Figure 5a). Figure 5d clearly shows that the $V^+/Ni^+$ ratios increased with the concentrations of $SO_4^{2-}$ and LO-OOA during the sea breeze from southwest direction. These observations confirm that a part of the $SO_4^{2-}$ and LO-OOA encountered on site were influenced by ship emissions and/or heavy oil combustion from the industrial zone.

### 3.5.2 Potential origins of MO-OOA

Although a large local source of biomass burning in Singapore is uncommon, it is important to investigate the potential influences of biomass burning events, including agricultural burning and forest fires, from the nearby countries (Figure S12a). Budisulistiorini et al. (2018) reports high mass loadings of biomass burning OA (BBOA) during the Indonesian wildfire in 2015 which had strong impacts on the air quality in the Southeast Asia. BBOA is not identified in the PMF analysis in this field study. However, the average fraction of $C_2H_4O_2^+$ to total organic ($f_{C2H4O2+}$) of 0.8% (Figure S13a) with ~86% of the data giving $f_{C2H4O2+}$ above 0.3% (i.e., a background $f_{C2H4O2+}$ values for non-biomass burning (Cubison et al., 2011)) in the laser-off mode measurement suggests potential influences from aged biomass burning emissions that cannot be easily separated by the conventional PMF analysis of organic fragments alone. Note that MO-OOA and COA present slightly higher $f_{C2H4O2+}$ values of 0.5% and 0.8% respectively (Figure 6a). However, the origin of MO-OOA is highly uncertain compared to other OA factors identified in this study due to the fact that atmospheric aging can diminish the distinctive mass spectral features of POA in general (e.g., decreasing in $f_{C2H4O2+}$ for BBOA (Cubison et al., 2011) and converging in the $f_{43}$-$f_{44}$ framework for OOA production (Ng et al. 2010)). Therefore, the following discussion will focus on evaluating the potential connections between the MO-OOA component and aged regional emissions.

Figures 6c and S14b show that high $Rb^+$ and $K^+$ signals were associated with more oxygenated ($f_{44} > 0.7$) fraction of OA, and moderate correlations between MO-OOA and the two metal were observed ($Rb^+$, r = 0.58 and $K^+$, r = 0.71, Figure S13b). Furthermore, the results of PMF analysis demonstrated that both $K^+$ and $Rb^+$ were mainly associated with MO-OOA (58-66%, Figure 5a) followed by the two combustion-related components (HOA and O-HOA). Although potassium and rubidium are not unique tracers for a specific combustion source, previous studies have shown that these two metals can be largely associated with biomass burning emissions (Artaxo et al., 1993; Lee et al., 2016; Achad et al., 2018). Note that rubidium has also been used as a coal combustion tracer in previous studies (Fine et al., 2004; Irei et al., 2014). Unlike ambient OA component, the chemical identities of $K^+$ and $Rb^+$ are unlikely modified by the oxidative aging of aerosol particles. Therefore, a strong temporal correlation between $Rb^+$ and $K^+$ (r = 0.85, Figure 6b) further suggests that they were likely of similar origins in this study.

The regional origin of $K^+$, $Rb^+$, $C_2H_4O_2^+$ and MO-OOA were investigated through their PSCF. Their PSCF graphs (Figures 6d and S15a- c) show several common origins with high probability that the highest concentrations could be influenced by biomass burning events from Indonesia (Figure S12a). Nevertheless, coal-fired power plants are located nearby the identified hotspots of $Rb^+$ and $K^+$ (Figure S12b) so that a regional transport of coal-fired power plant emissions alongside biomass burning plumes were possible. Note that MO-OOA contributes to the highest fraction of nitrogen-containing organic fragments ($C_xH_yNO_z^+$ ~ 32% and $C_xH_yN^+$ ~ 46%) that can be generated by biomass burning emissions (Mace et al., 2003; Laskin et al., 2009; Desyaterik et al., 2013; Mohr et al., 2013). It is important to point out that most of the previous studies usually describe MO-OOA (or LV-OOA in some earlier studies) as aged SOA component without providing further detail on their potential origin and emission characteristics. Our observations underline the possibility of better understanding the origin of the MO-OOA component through measurements of refractory metals even when atmospheric oxidative processing has made the mass spectral features of aged OA materials less distinguishable.

## 4.    Conclusions and Atmospheric Implications

Real-time aerosol mass spectrometry techniques have been successfully deployed in the fields worldwide in the last two decades and their observations have substantially enhanced our quantitative understanding of sources and formation of NR-PM, in particular through the use of PMF analysis for OA source apportionment (Fröhlich et al., 2015; Ng et al., 2010; Zhang et al., 2007b). In this work, we demonstrate the use of refractory aerosol components, rBC and a few trace metals, measured by an Aerodyne SP-AMS to better characterize the sources, emission characteristics and aging history of ambient OA and rBC in Singapore, a highly urbanized tropical city that is influenced by multiple local and regional air pollution sources.

By integrating rBC fragment ($C_n^+$) into the PMF analysis, we can identify the major sources and characteristics of ambient rBC and OA particles based on their degree of associations. In particular, the majority of total rBC (> 90%) was originated from local combustion emissions, in which ~30% of them were largely associated with the fresh SOA components (LO-OOA). This observation implies that a fraction of rBC could act as condensation sinks of fresh SOA in this field study although additional information is required to determine the mixing state of rBC and SOA. We further illustrate the potential application of relative intensities of major rBC fragments for aerosol source identification. While higher $C_1^+/C_3^+$ ratios (> 0.7) were associated with rBC originated from the industrial zone and shipping port (O-HOA and LO-OOA), rBC associated with traffic (HOA) gave lower $C_1^+/C_3^+$ ratios (~0.65). Trace metals analysis further shows that the high $C_1^+/C_3^+$ ratios coincided with high $V^+/Ni^+$ ratios, highlighting the potential influences of emission from shipping activities and oil refinery industry on the chemical characteristics of O-HOA and LO-OOA. The observed correlation between $Na^+$ and rBC suggests that the contribution of anthropogenic emissions to total particulate sodium should not be ignored in the coastal urban environments. These results underline the advantages of using refractory aerosol component to disentangle various combustion sources encountered in an urban environment influenced by multiple sources using a single online instrument.

One of the major challenges in interpreting the PMF results from AMS measurements is to identify the origin and aging history of ambient particles associated with highly oxidized OA components (e.g., MO-OOA in this work), as the mass spectral characteristics of OA converge along with their degree of oxidative aging. In general, the relative intensities of some highly oxygenated fragments (e.g. $CO^+$ and $CO_2^+$) increase continuously while other mass spectral features (e.g. alkane/alkene patterns from combustion sources) being diminished during the aging processes. Ng et al. (2010) have visualized such phenomena for SOA components on the $f_{43}$-$f_{44}$ space. Furthermore, the $m/z$ 60 (or $C_2H_4O_2^+$ organic fragments) has been widely used as a tracer ions for BBOA. Cubison et al. (2011) have generalized observations from worldwide field data that $m/z$ 60 signature become less and less significant in aged BBOA materials. In this study, we proposed that the MO-OOA component represented aged OA materials impacted by the regional biomass burning and perhaps coal combustion emissions as MO-OOA was associated with refractory $K^+$ and $Rb^+$. Given the fact that MO-OOA was the major OA components of the total OA (~32%), this result highlights the fact that the regional pollution can affect the air quality in Singapore even though fresh regional biomass burning episodes were not observed during the sampling period.

More broadly, the improved source identification for OA and rBC can provide useful information to further investigate the effects of atmospheric aging on their physio-chemical properties. For example, this work highlights

the potential influences of regional biomass burning and coal combustion emissions to MO-OOA component, which may provide important insight into how light absorbing properties of OA (i.e., brown carbon) evolve with transport and aging. Recently, Dasari et al. (2019) provided field evidence for the bleaching of brown carbon during their transport by photo-oxidation and that photo-dissociation can occur in the South Asian outflow based on measurements near and away from specific combustion sources, including biomass burning and traffic. Furthermore, this type of PMF analysis could be applied to analyze the sources and characteristics of rBC-containing particles exclusively (i.e., rBC core with organic coatings) in order to advance our understanding on the effects of primary emissions and/or atmospheric processing on BC light absorption enhancement caused by the lensing effect.

**Acknowledgment:** This work is supported by the National Environmental Agency (NEA) of Singapore (NEA, R-706-000-043-490) and by the National University of Singapore start-up grant (R-302-000-173-133). The content does not represent NEA's view.

**Author contributions:** A.K.Y.L. supervised the project. M.G.A. and N.K. carried out the experiment. L.-H.R. and M.G.A analysed the data. L.-H.R. wrote the manuscript with support and comments from all the co-authors.

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

Table 1: Top: Carbon fragment ratios observed by the laser-on measurement for each PMF factor. The underlined number represents the results with both rBC fragments and trace metal ions included as PMF input. Bottom: Contribution of each PMF factor to the total signal of specific metal ions. For the entire table, the values in parenthesis are the one obtained from PMF with CDCE-corrected input matrix (see details in Text S1 and Table S4 of the supplementary information).

| PMF factors | HOA | O-HOA | COA | LO-OOA | MO-OOA |
|---|---|---|---|---|---|
| Carbon fragments ratios | | | | | |
| $C_1^+/C_3^+$ | 0.66 (0.63) <br> 0.65 (0.62) | 1.00 (0.90) <br> 1.00 (0.89) | NA* | 0.81 (0.88) <br> 0.79 (0.85) | NA# |
| $C_2^+/C_3^+$ | 0.38 (0.39) <br> 0.38 (0.39) | 0.41 (0.40) <br> 0.41 (0.40) | NA* | 0.39 (0.41) <br> 0.41 (0.40) | NA# |
| Contribution of each factor to the total signal of specific metal ions (fraction) | | | | | |
| $Na^+$ | 0.35 (0.22) | 0.14 (0.17) | < 0.01 (< 0.01) | 0.45 (0.58) | 0.06 (0.03) |
| $K^+$ | 0.23 (0.23) | 0.19 (0.18) | < 0.01 (< 0.01) | <0.01 (0.05) | 0.58 (0.54) |
| $V^+$ | 0.21 (0.08) | 0.09 (0.16) | < 0.01 (< 0.01) | 0.70 (0.76) | < 0.01 (< 0.01) |
| $Ni^+$ | 0.38 (0.22) | 0.20 (0.22) | < 0.01 (< 0.01) | 0.29 (0.45) | 0.13 (0.11) |
| $Rb^+$ | 0.15 (0.15) | 0.19 (0.19) | < 0.01 (< 0.01) | <0.01 (0.01) | 0.66 (0.65) |

* None of the refractory $C_n^+$ fragments were associated with the COA factor
# Large variations of $C_n^+$ fragment ratios between cases, hence the ratios were not reported for MO-OOA

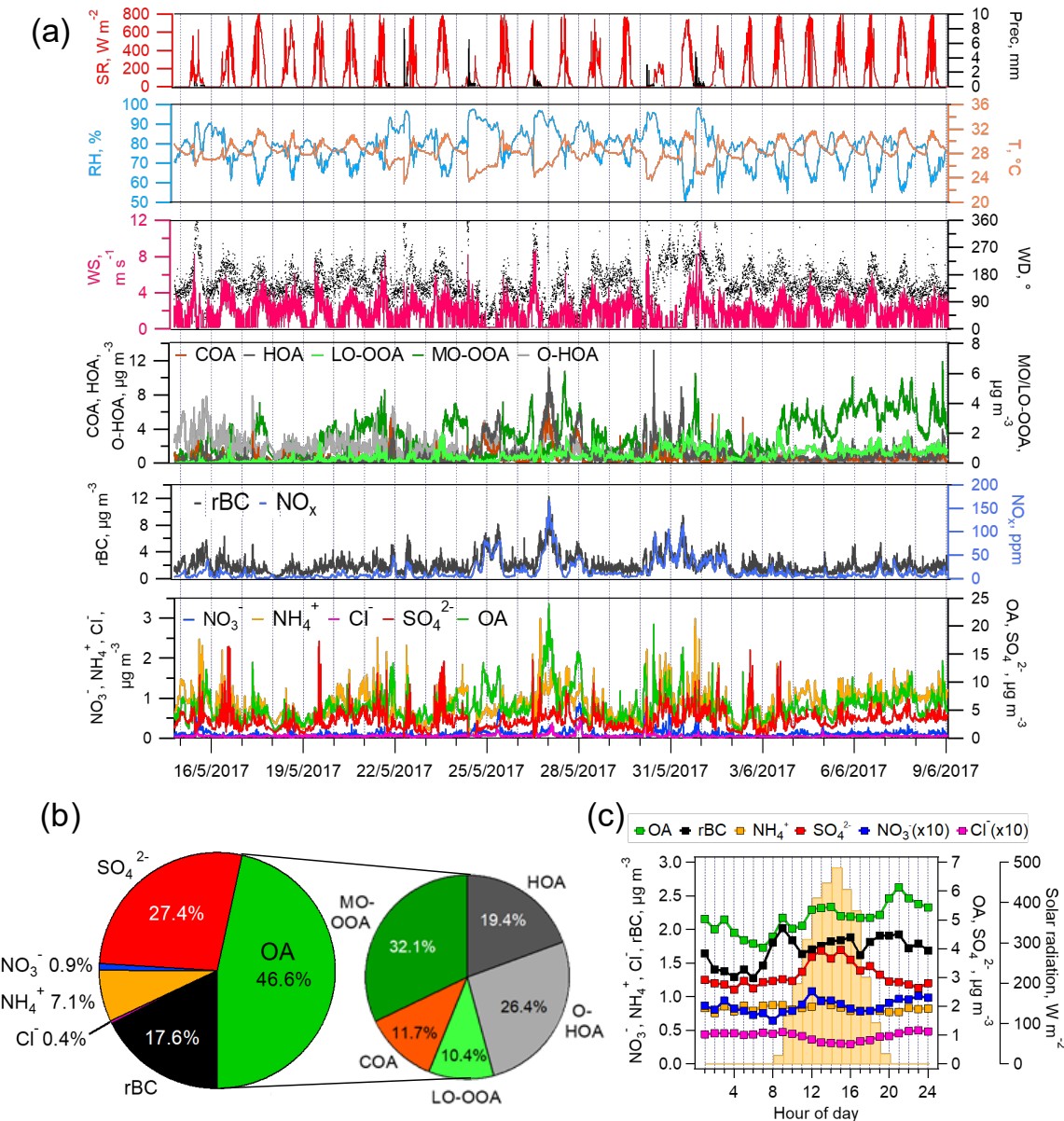

Figure 1: (a) Time series of solar radiation (SR), precipitation, temperature, relative humidity, wind speed (WS) and direction (WD), PMF factors (HOA, O-HOA, COA, LO-OOA, and MO-OOA from laser-off measurements), rBC, $NO_X$, total OA and inorganic species. (b) Average chemical compositions of NR-PM and rBC with the contribution of the PMF factors to total OA. (c) Diurnal patterns of rBC, total OA, inorganic species and solar radiations (represented by the yellow bars).

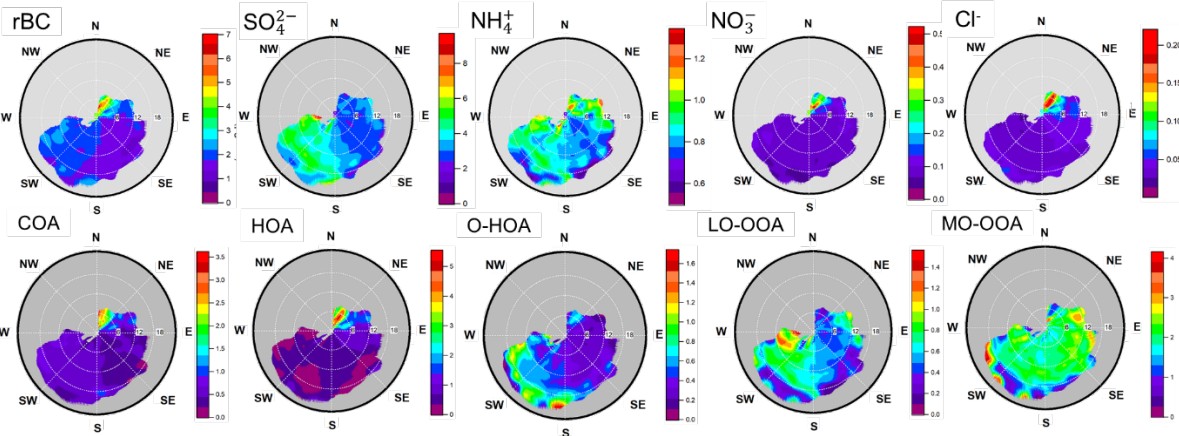

Figure 2: NWR plots of aerosol components measured by the SP-AMS during the whole campaign (Note: all species and OA components are from laser-off measurements except for rBC).

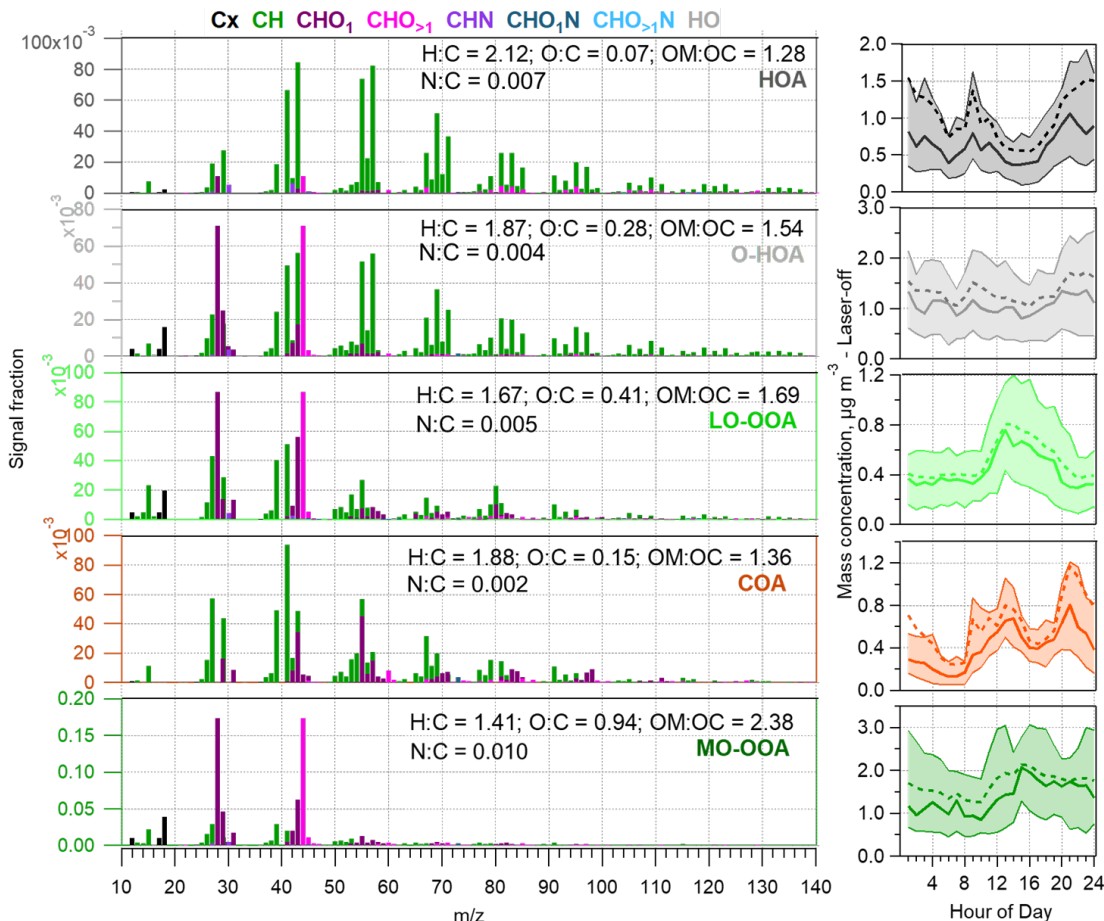

Figure 3: (Left) Mass spectra and elemental ratios of the five-factor solution obtained from the PMF analysis of laser-off measurements and (right) their respective diurnal cycles (median - plain line, mean – dotted line, 25th and 75th centiles - shaded area) for the entire campaign. The PMF results from laser-on measurements are shown in Figure S4.

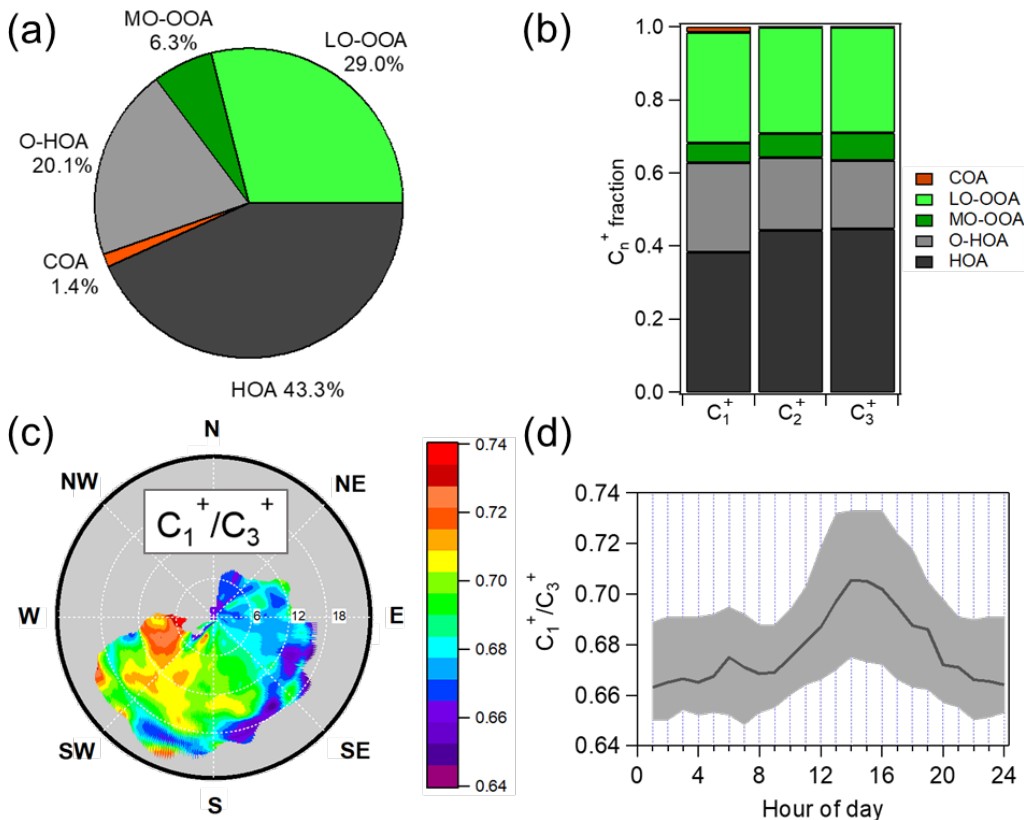

Figure 4: (a) Fractions of rBC and (b) relative contributions of $C_1^+$-$C_3^+$ fragments contributed by the five OA components identified by the PMF analysis with OA and rBC fragments measured by the laser-on mode as model inputs. (c) NWR plot and (d) diurnal cycle of $C_1^+/C_3^+$ ratio (25th and 75th centiles - shaded area).

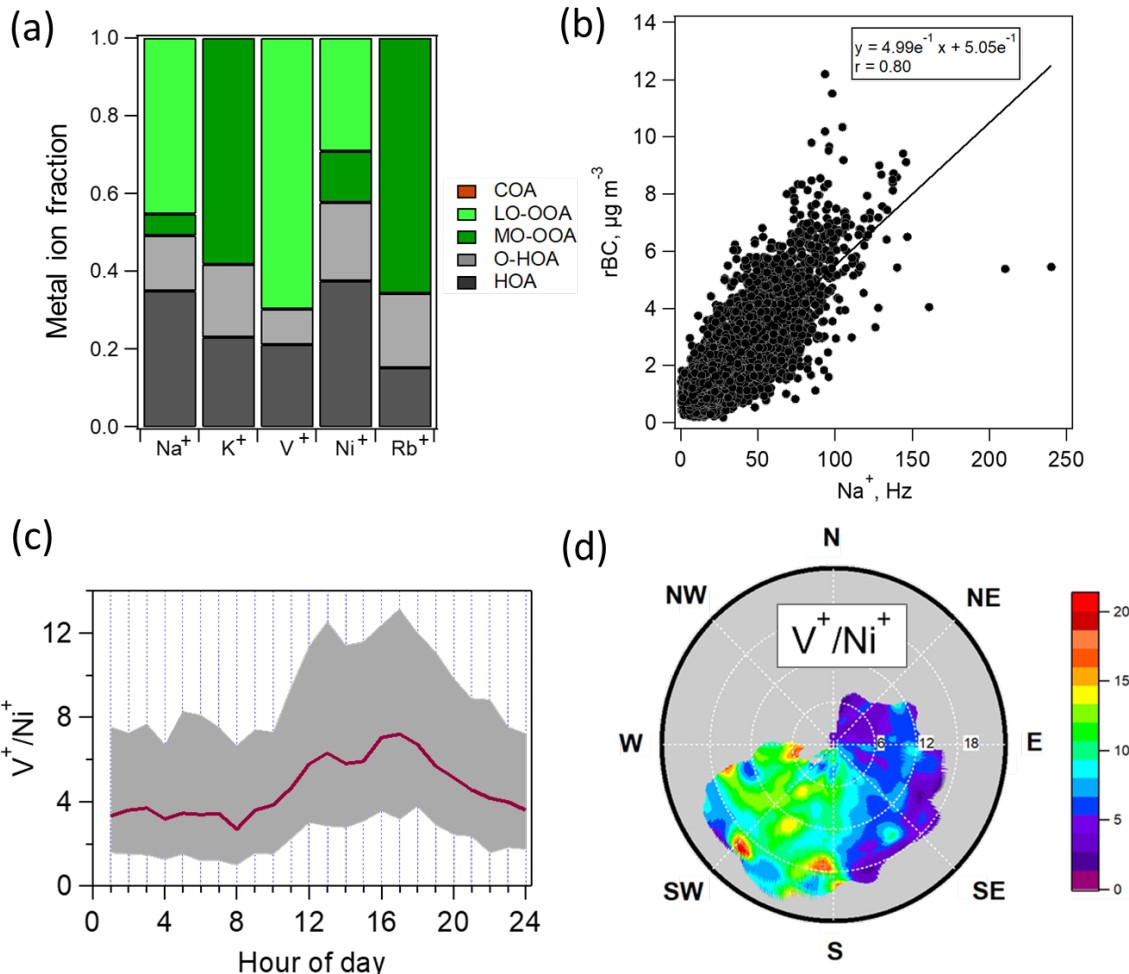

Figure 5: (a) Fractions of trace metal ions contributed by each OA component obtained from the PMF analysis with OA, rBC, and metal ions measured by the laser-on mode as model inputs. (b) Scatter plot of 5-min averaged rBC concentrations vs. Na$^+$ signals. (c) Diurnal cycle of V$^+$/Ni$^+$ ratio (25[th] and 75[th] centiles - shaded area). (d) NWR plot of V$^+$/Ni$^+$ ratio over the entire campaign.

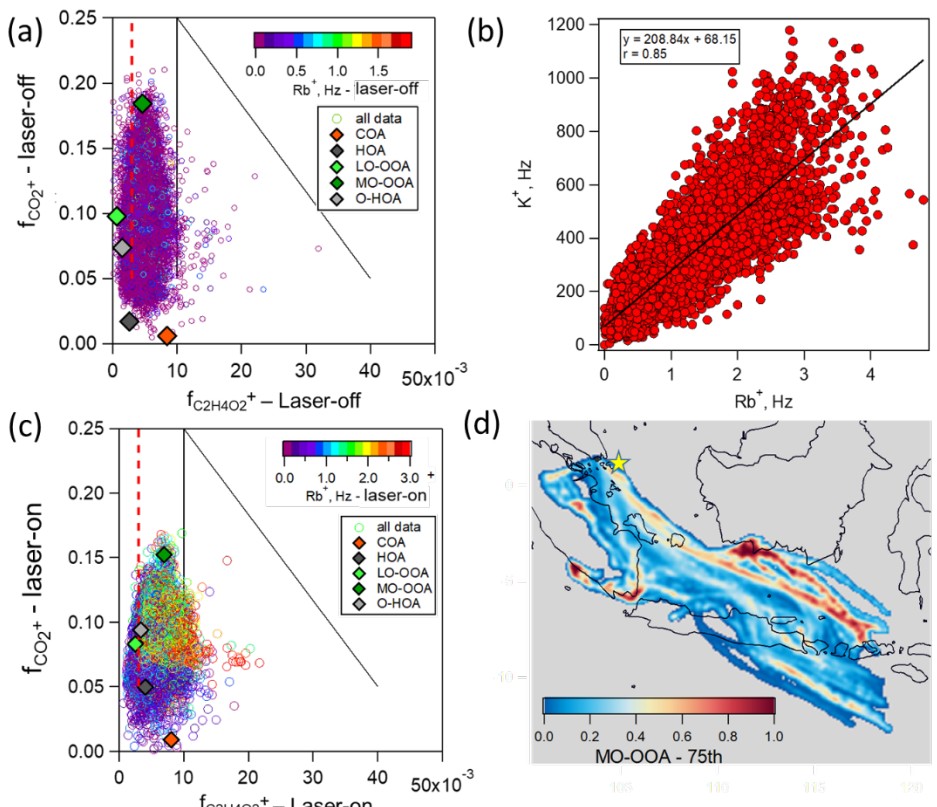

Figure 6: Scatter plots of $f_{CO2+}$ vs. $f_{C2H4O2+}$ with the symbol color scaled by the rubidium ($Rb^+$) ion signals based on (a) laser-off and (c) laser-on measurements (Dash red line – 0.3% background value, plain black lines define the space with and without BB influence (inside and outside the triangular region, respectively, Cubison et al., (2011)). (b) Scatter plot of potassium ion ($K^+$) vs. rubidium ($Rb^+$) signals from the laser-on measurements. (d) PSCF graph of MO-OOA by considering only pollutant concentrations above their respective 75th percentile (Figure S15 shows PSCF graphs of $K^+$, $Rb^+$, and $C_2H_4O_2^+$)