# Peer review of "Characterization of carbonaceous aerosols in Singapore: insight from black carbon fragments and trace metal ions detected by a soot-particle aerosol mass spectrometer"

_Atmospheric Chemistry and Physics, 2019_

## Referee Comment (RC1) · Anonymous Referee #1 · 13 Nov 2019

The manuscript "Characterization of carbonaceous aerosols in Singapore: insight from black carbon fragments and trace metal ions detected by a soot-particle aerosol mass spectrometer" investigated the sources of carbonaceous aerosol in Singapore by using Positive Matrix Factorization. Besides utilizing the mass spectra of organics, they inserted the mass fragments of rBC and metals in the data matrix in order to improve the interpretation of the aerosol sources in urban area. The authors were able to resolve five sources/types of organic aerosols; hydrocarbon-like OA, oxygenated-HOA, cooking-related OA, less-oxidized oxygenated OA and more-oxidized oxygenated OA.

This paper presents interesting results and novel data analysis methods. Paper is well-written and fluent, and in most parts also easy to understand. It utilizes nicely the whole high resolution mass spectra provided by the SP-AMS, and thoroughly validates the PMF results. However, the interpretation of the PMF factors is sometimes confusing. Additionally, there are shortcomings in the measurement methods, and the authors have drawn some conclusion based on rather weak evidences. Therefore I recommend publication of the manuscript only after the comments below are adequately addressed.

General comments

The quantitativity of the SP-AMS results is not considered adequately. Since there are several sources of uncertainty in the AMS measurement, e.g. CE, IE and RIE values, the validation of the SP-AMS needs to be done more accurately. Additionally, the weakness of this paper is that only the data from the SP-AMS is utilized. I assume there were also other instruments at the site (for PM, size distribution, BC etc.), or at the air quality monitoring stations nearby, that could be used for the validation/comparison of the SP-AMS data. Without auxiliary data, there accuracy of the given concentrations remains vague.

Regarding PMF, I don't understand why only few five metal ions were included in the PMF model even though the SP-AMS can detect quite a few metals. For example, Al, Ca, Cd, Cr, Cu, Fe, Mn, Zn and Sn could be included. I would assume that the inclusion of more metals would improve the interpretation of PMF results. I suggest adding more metals to the PMF analysis because I think that all the interesting information in the data set is not used so far.

In terms of metals, I disagree with the authors that all of the metals presented here can be used as tracers. I agree that Na can come from fuel but it can also originate from other sources in urban areas, for example from coal and biomass burning (e.g. Hsu et al. 2011). Similarly, Rb signal in AMS can originate from coal combustion (Irei et

al., 2014). I suggest that the authors consider other sources of metals in urban areas (especially close to industrial areas), and revise their conclusion regarding metals.

Specific comments

1. Page 1, Abstract; lines 19-20; "local combustion sources" and "industrial emissions" need to be described in more detailed. What kind of combustion sources, just traffic? What industry there are in that area?

2. Page 4, line 27; "composition –dependent approach" CE values need to be shown in the paper and the uncertainty of the CE has to be evaluated carefully. Since the RIE for sulfate was not determined, the uncertainty analysis is very important.

3. Page 5, line 13; I don't understand why only few five metal ions were included in the PMF model. I suggest adding more metal ions.

4. Page 5, line 13-14; metals ions were included in Hz's into the PMF. How does that affect the results of the PMF?

5. Page 5, line 30; sulfate increase starting from ∼10:00 can be explained by the active photochemistry. Could you add a diurnal plot showing solar radiation and sulfate in the same figure?

6. Page 5, line 30; NH4 ratio and acidity; the discussion on acidity is somewhat unreliable because the RIE for sulfate was not measured. For example in Fig. S4a, it's hard to understand why all the points are below 1:1 line. To me, this seems to be due to the incorrect RIE for sulfate. Could you comment on that?

7. Page 7, line 5; industrial emission; be more specific what kind of emissions

8. Page 7, line 28; define POA, is it only HOA? Isn't COA also primary?

9. Page 7, line 38; "large combustion sources during nighttime" needs to be elaborated since it has such a large effect on diurnal cycle of HOA

10. Page 9, lines 11-15; "organo-nitrate and nitrogen-containing fragments increased during the daytime in this field study." Why there is not more discussion on these species if they were detected? What is their contribution to OA or LO-OOA? Add the ratio of N to C (and S:C) to Figure 3.

11. Page 11, line 5-; Na as a tracer for fossil fuel combustion. Revise this section by taking into account other sources of Na in urban areas (see general comments).

12. Page 11, Biomass burning tracers, I don't understand why the contribution of biomass burning has been investigated so extensively even though it does not show up in PMF and the metals are not very good tracers for it. Consider making Section 3.5.2. more compact.

13. Page 12, line 24-26; "Kasthuriarachchi et al. (2019) also found that MO-OOA was the major contributor to the observed nitrogen-containing organic fragments that could be largely generated by biomass burning emissions" is contradictory to page 9: "Kasthuriarachchi et al. (2019) reported that the concentrations of organo-nitrate and nitrogen-containing fragments (i.e., CxHyN+ and CxHyNOz+) increased during the daytime in this field study, suggesting that photo oxidation of VOCs under high-NOX condition could be another potential pathway toward LO-OOA formation". Please revise the sentence/sentences

14. Page 12, line 31-33 and Fig. S11, Fig. S11 is confusing. It is very difficult to see which trajectories really pass the two fires shown by the MODIS. Could you just select the trajectory for the peak concentrations of m/z 60? Where are the large urban areas in the map? Do the trajectories pass large cities as well? Is it possible that biomass burning emissions were mixed with urban emissions (that increased Rb and maybe also K)? Please add cities to the map. Why Fig. S11(b) and Fig S11(c) are not on the same scale? Now it is difficult to compare (b) and (c). Add lines between markers to m/z 60 and Rb in Fig. S11a.

15. Page 13, lines 21-23; "Furthermore, Na+ measured by the SP-AMS can potentially

be useful for separating rBC from traffic and biomass burning combustion emissions, which required further investigation." I suggest revising this sentence as the evidences provided by this study are not convincing enough. I think that there are multiple sources for Na in urban areas that should be taken into account.

16. Page 13, lines 36-37; "This observation suggest that, in the region, both K+ and Rb+ might be more appropriate tracers for the identification of aged biomass burning than m/z 60." I suggest revising this sentence as Rb may also come from other sources.

Technical corrections:

1. Page 7, line 13; less-oxidized oxygenated OA (LO-OOA)

2. Supplemental material, could you add a figure showing the contribution of metals to each PMF factor? It would be interesting to see how metals divide between the factors.

References

Hsu, L. J., Alwahabi, Z. T., Nathan, G. J., Li, Y., Z. LI, S., Aldén, M.(2011). Sodium and Potassium Released from Burning Particles of Brown Coal and Pine Wood in a Laminar Premixed Methane Flame Using Quantitative Laser-Induced Breakdown Spectroscopy. Applied spectroscopy, 65, 684-91.

Irei, S., Shimono, A., Hikida, T., Kuramoto, K., Suzuki, Y., Takami, A. (2014). Qualitative Evaluation of m/z 85, 87, and 133 Signals in Organic Aerosol Mass Spectra of Fly Ash Produced by Coal Combustion. Aerosol and Air Quality Research, 14: 406–412.

---

## Referee Comment (RC2) · Eleanor Browne (Referee) · 29 Dec 2019

Rivellini et al made ambient measurements of aerosol chemical composition with an SP-AMS on the campus of the National University of Singapore. PMF analysis was performed on the data and the authors investigated how the inclusion of rBC and metals into the PMF affected the interpretation of the PMF results. Overall, this is an interesting data set that highlights how rBC and metal measurements can enhance our understanding of aerosol sources.

Major comments 1) The SP-AMS calibrations and data analysis need to be described in more detail. In particular, CE needs to be considered/discussed in more detail. Is there evidence for the morphology and coating of the rBC particles varying at all over the course of the measurements (and consequently a varying CE)? How would a varying CE affect the interpretation of the measurements and the PMF results?

2) In Sect. 3.4, the C1/C3 analysis should include a discussion about how uncertainty in the contribution of C1+ from non-refractory organic aerosol affects the interpretation of the ratio, particularly since a constant C1/C3 ratio was used to as a correction factor (Sect. 2.2). It may be beneficial to investigate the ratio of other Cx ions (e.g., C4/C3) to provide insight into the possible contribution of organics to C1.

3) Sect. 3.5.1 would be strengthened by a more detailed discussion regarding urban sources of metals and including more metals in the analysis. Additionally, the potential contribution of oceanic sources to Na+ should be discussed in more detail. While the authors state that Na+ and Cl- exhibit poor temporal correlation, I assume this is for the whole data set. If one filters by wind direction (and thus sea breeze) does this still hold? Additionally, is there an interpretation for the large contribution of Na+ in LO-OOA? Finally, does the fact that different C1/C3 ratios were seen for LO-OOA and HOA (the factors with the strongest Na+ contribution) tell us anything about the different sources of Na+?

4) To me, it seems that the last sentence of the introduction and the last paragraph of the conclusions overstate the implications and applications of the manuscript. Optical properties of aerosols were not discussed in the manuscript, so the inclusion in the introduction and conclusion seems out of place. Moreover, the work of Kasthuri-arachchi et al (2019) is under review so the inclusion here is somewhat misleading. I suggest that the authors consider revising this paragraph and the last sentence of the introduction to better reflect the material covered in this study.

Minor Comments -It would be helpful to explicitly pinpoint some of the main industrial

sources (such as the oil refinery discussed in the introduction and shipping ports) in Figure S1a.

-Page 7 lines 12-15: Regarding the production of organo-sulfur compounds in acidic sulfate plumes – this seems rather poorly supported at the moment. As it does not advance the main focus of the paper, I suggest removing.

-Figure S4a the legend (colors based on CH2SO) doesn't match the caption (data originating from SW and other directions).

-Figure S4d needs a color scale. The 2 modes (discussed at the end of page 6) are also not apparent from this figure, perhaps due to the color scale used. Currently it looks to me more like the distribution broadens when the concentration increases in the middle of the day.

-Discuss figures in order (e.g., Fig. S6 is currently referenced before S5). Cite panels of figures in order (for instance, the panels of Fig. S4 are cited out of order).

-Figure S5: The wording in the caption is unclear, specifically "... both filtered from 25th to 28th of May, over the campaign." Is "both" just COA and HOA? Does it mean that only the 25th to 28th was included?

-In Sect 3.5.2, why use m/z 60 rather than the HR ion (C2H4O2+)?

-Page 10 line 27: rather than "inferior" use "smaller than"

-Page 12 line 23: I do not find the similar size distributions for m/z 39 and 60 to be convincing evidence for a BBOA source. m/z 60 is the only individual organic ion shown in Fig. S8b. How unique is the size distribution for m/z 60 compared to other individual organic ions?

-Page 12 line 19: should this be Figure 5d rather than 3c and 4b?

---

## Author Comment (AC2) · 22 Mar 2020

The comment was uploaded in the form of a supplement:
https://www.atmos-chem-phys-discuss.net/acp-2019-857/acp-2019-857-AC2-supplement.pdf

---

## Author Response (AR1)

The manuscript "Characterization of carbonaceous aerosols in Singapore: insight from black carbon fragments and trace metal ions detected by a soot-particle aerosol mass spectrometer" investigated the sources of carbonaceous aerosol in Singapore by using Positive Matrix Factorization. Besides utilizing the mass spectra of organics, they inserted the mass fragments of rBC and metals in the data matrix in order to improve the interpretation of the aerosol sources in urban area. The authors were able to resolve five sources/types of organic aerosols; hydrocarbon-like OA, oxygenated-HOA, cooking-related OA, less-oxidized oxygenated OA and more-oxidized oxygenated OA.

This paper presents interesting results and novel data analysis methods. Paper is well-written and fluent, and in most parts also easy to understand. It utilizes nicely the whole high resolution mass spectra provided by the SP-AMS, and thoroughly validates the PMF results. However, the interpretation of the PMF factors is sometimes confusing. Additionally, there are shortcomings in the measurement methods, and the authors have drawn some conclusion based on rather weak evidences. Therefore I recommend publication of the manuscript only after the comments below are adequately addressed.

Author's response: We thank for the positive comments from the reviewer. We have provided our point-to-point responses below (in blue colour) to address the questions/comments as shown below:

**General comments**
The quantitativity of the SP-AMS results is not considered adequately. Since there are several sources of uncertainty in the AMS measurement, e.g. CE, IE and RIE values, the validation of the SP-AMS needs to be done more accurately. Additionally, the weakness of this paper is that only the data from the SP-AMS is utilized. I assume there were also other instruments at the site (for PM, size distribution, BC etc.), or at the air quality monitoring stations nearby, that could be used for the validation/comparison of the SP-AMS data. Without auxiliary data, there accuracy of the given concentrations remains vague.

Author's response: Thanks for the reviewer's suggestion for addressing the SP-AMS measurement uncertainties. In this field study, we conducted four calibrations and obtained an average $IE_{NO3}$ (ionization efficiency of nitrate) value of 7.92e-8 ($\pm$ 4.08e-09). The standard deviations of RIEs (relative ionization efficiency) for $NH_4^+$ and rBC have been added in the main text.

The SP-AMS measurements were compared with other co-located measurements from an aethalometer (model AE33, Magee scientific), an OC/EC analyser (Sunset Laboratory) and a Monitor for AeRosols and GAses in ambient air analyzer (MARGA, Metrohm) over the sampling period. Sulfate ($SO_4^{2-}$) and organic mass concentrations from the SP-AMS (CDCE-corrected) were compared to $SO_4^{2-}$ and OM measured by the MARGA and the OC/EC analyser, respectively (Figure S3a and b in the revised SI). Aside from the good correlations that underline a good temporal response from all instruments, the slopes are in the range of 0.81-0.88, suggesting that the mass concentrations of $SO_4^{2-}$ and organic measured by the SP-AMS are ~12-19% lower than those measured by the MARGA and the OC/EC analyser, respectively. This could be partially explained by the 1 µm cut-size of the SP-AMS, while the two other instruments were measuring $PM_{2.5}$.

The rBC mass concentrations (CE = 0.6) were compared with BC and EC measured by the aethalometer and the OC-EC analyser, respectively (Figure S3c and d in the revised SI). Scatter plots of those comparisons show Pearson coefficients > 0.84, and respective slopes of 0.83 and 1.10, supporting that CE = 0.6 for rBC is a reasonable CE correction approach. Note that no standard approach has been developed for determining CDCE of rBC measured by SP-AMS. As shown in our response to the second reviewer, we determined the CDCE of rBC by comparing BC and EC measured by the aethalometer and the OC/EC analyser, respectively. The median CDCE values ranged between 0.52

and 0.69 as shown in Figure S2 of the revised SI. The following sentences and Figure S3 have been added to the revised manuscript and SI based on the above observations:

Page 2 lines 7-9: "Other co-located instruments for $PM_{2.5}$ characterizations include an aethalometer (AE33, Magee Scientific), a Monitor for Aerosols and Gases (MARGA, Metrohm) and a semi-continuous organic and elemental carbon (OC/EC) analyser (Sunset Laboratory)."

Page 4 lines 27-29: "The campaign averages of $RIE_{rBC}$ and $RIE_{NH4}$ were 0.15 (±0.04) and 4.24 (±0.04), respectively. The default RIE values of nitrate (1.1), sulfate (1.2) and organics (1.4) were used for respective mass concentration quantification (Jimenez, 2003)."

Page 4 lines 33-38: "The SP-AMS measurements were compared with the sulfate and OM concentrations measured by the MARGA and the OC/EC analyser, respectively, showing strong temporal (r = 0.77 and 0.93) and quantitative (slopes = 0.81 and 0.88) agreements between these measurements (Figure S3a and b). A collection efficiency (CE) of 0.6 was used for rBC quantification due to incomplete overlap between the laser vaporizer and the particle beam (Willis et al., 2014). The corrected rBC concentrations were also comparable to those measured by the aethalometer (r = 0.96, slope = 0.83) and the EC measured by the OC/EC analyser (r = 0.84, slope = 1.1) (Figure S3c and d)."

[Figure]

Figure S3: Scatter plots of (a) the hourly-averaged $SO_4^{2-}$ mass concentration measured by the SP-AMS and MARGA, (b) and the hourly-averaged OA and OM (estimated from OC concentration using a 2.2 conversion factor) mass concentrations measured by the SP-AMS and OC/EC analyser, respectively. Comparisons of 10-min averaged rBC mass concentrations measured by the SP-AMS with (c) BC and (d) EC mass concentrations measured by the aethalometer and OC/EC analyser, respectively.

Regarding PMF, I don't understand why only few five metal ions were included in the PMF model even though the SP-AMS can detect quite a few metals. For example, Al, Ca, Cd, Cr, Cu, Fe, Mn, Zn and Sn could be included. I would assume that the inclusion of more metals would improve the interpretation of PMF results. I suggest adding more metals to the PMF analysis because I think that all the interesting information in the data set is not used so far.

Author's response: We agree with the reviewer that it is important to include all the detectable metal signals in the PMF analysis if possible. However, we only reported metals with their average values higher than the limit of detection (Table S2) determined by the particle-free air to ensure the quality of our data analysis. Nickel is a marginal case but a large fraction of its signal remain above its detection limit for investigating the $V^+/Ni^+$ ratio. To make this point clear, we have revised the sentence in Section 2.3.1 as shown below.

Page 5 lines 24-25: "Lastly, five metal ions ($K^+$, $Na^+$, $Ni^+$, $V^+$, $Rb^+$) were included into the PMF model (Figure S4g-i) as the majority of their signals were higher than their respective limit of detection (Table S2)."

In terms of metals, I disagree with the authors that all of the metals presented here can be used as tracers. I agree that Na can come from fuel but it can also originate from other sources in urban areas, for example from coal and biomass burning (e.g. Hsu et al. 2011). Similarly, Rb signal in AMS can originate from coal combustion (Irei et al., 2014). I suggest that the authors consider other sources of metals in urban areas (especially close to industrial areas), and revise their conclusion regarding metals.

Author's response: We agree with the reviewer that most of the metal species that we discussed in the manuscript are not unique tracers. Therefore, our intention is not to discuss how those trace metal species can be used to specify the sources of OA. Rather, we understand that each OA factor identified by the PMF analysis in this work can be originated from different sources. To avoid confusion of our focus in Section 3.5, the sub-heading has been changed to "Characterization of metal ions associated with different OA sources".

With such information, we would like to improve our understanding on the emission characteristics of different OA factors and/or trace metals. For examples, we have pointed out in the manuscript that $Na^+$ was mainly associated with three OA factors (i.e., HOA, O-HOA, and LO-OOA), which reflects the fact that $Na^+$ could be emitted from different local combustion sources. Singapore has a biomass clean coal cogeneration plant, but it is important to highlight that natural gas is the major source of electricity (> 95%) in Singapore. Although we didn't observe BBOA factor in this study, we have mentioned that biomass burning can be a possible source of $Na^+$ in the revised version as suggested by the reviewer. A few sentences in Section 3.5.1 have been revised or added as shown below:

Page 12 lines 5-7: "Based on the PMF results that includes trace metal ions, sodium ($Na^+$) was mainly associated with HOA, O-HOA and LO-OOA (Figure 5a) that could be due to different types of fossil fuel combustion emissions (e.g., local traffic, shipping, and various industrial activities) as discussed in Section 3.4."

Page 12 lines 20-23: "Biomass burning can be a possible source of $Na^+$ (Hsu et al., 2011) but no major fresh biomass burning emissions were observed in this study. The MO-OOA factor is suspected to be more influenced by aged regional biomass burning emissions (see more discussion in Section 3.5.2) but $Na^+$ was not strongly associated with this factor."

Similar to $Na^+$, we understand that $Rb^+$ is not a unique tracer for biomass burning. Rather, $Rb^+$ is likely associated with multiple sources. The results of PMF analysis demonstrated that both $K^+$ and $Rb^+$ were mainly associated with MO-OOA (58-66%) followed by the two combustion-related components (HOA and O-HOA). As suggested by the reviewer, we have mentioned that $Rb^+$ can be a tracer of coal

combustion based on previous studies. Although local coal combustion is uncommon in Singapore, the regional transported of coal combustion plant emissions were still possible. In the revised manuscript, we have revised the discussion to highlight that both regional biomass burning and coal combustion emissions are possible sources of $Rb^+$. However, it won't substantially change our conclusion that trace metal can allow us to better understand the potential origins of MO-OOA. Since MO-OOA usually represents aged SOA materials in many previous field studies and is a PMF OA factor that cannot provide much mass spectral characteristic for source identification, our data interpretation can provide insight into better identification of MO-OOA sources and/or aging history for other SP-AMS measurements in the future. The majority of Section of 3.5.2 has been rewritten as following.

Page 13, line 1: The sub-heading has been changed to "Potential origins of MO-OOA" to avoid confusion of using $Rb^+$ as a unique tracer for biomass burning.

Page 13, lines 16-37: "Figures 6c and S14b show that high $Rb^+$ and $K^+$ signals were associated with more oxygenated ($f_{44} > 0.7$) fraction of OA, and moderate correlations between MO-OOA and the two metal were observed ($Rb^+$, r = 0.58 and $K^+$, r = 0.71, Figure S13b). Furthermore, the results of PMF analysis demonstrated that both $K^+$ and $Rb^+$ were mainly associated with MO-OOA (58-66%, Figure 5a) followed by the two combustion-related components (HOA and O-HOA). Although potassium and rubidium are not unique tracers for a specific combustion source, previous studies have shown that these two metals can be largely associated with biomass burning emissions (Artaxo et al., 1993; Lee et al., 2016; Achad et al., 2018). Note that rubidium has also been used as a coal combustion tracer in previous studies (Fine et al., 2004; Irei et al., 2014). Unlike ambient OA component, the chemical identities of $K^+$ and $Rb^+$ are unlikely modified by the oxidative aging of aerosol particles. Therefore, a strong temporal correlation between $Rb^+$ and $K^+$ (r = 0.85, Figure 6b) further suggests that they were likely of similar origins in this study.

The regional origin of $K^+$, $Rb^+$, $C_2H_4O_2^+$ and MO-OOA were investigated through their PSCF. Their PSCF graphs (Figures 6d and S15a- c) show several common origins with high probability that the highest concentrations could be influenced by biomass burning events from Indonesia (Figure S12a). Nevertheless, coal-fired power plants are located nearby the identified hotspots of $Rb^+$ and $K^+$ (Figure S12b) so that a regional transport of coal-fired power plant emissions alongside biomass burning plumes were possible. Note that MO-OOA contributes to the highest fraction of nitrogen-containing organic fragments ($C_xH_yNO_z^+ \sim 32\%$ and $C_xH_yN^+ \sim 46\%$) that can be generated by biomass burning emissions (Mace et al., 2003; Laskin et al., 2009; Desyaterik et al., 2013; Mohr et al., 2013). It is important to point out that most of the previous studies usually describe MO-OOA (or LV-OOA in some earlier studies) as aged SOA component without providing further detail on their potential origin and emission characteristics. Our observations underline the possibility of better understanding the origin of the MO-OOA component through measurements of refractory metals even when atmospheric oxidative processing has made the mass spectral features of aged OA materials less distinguishable."

Specific comments
1. Page 1, Abstract; lines 19-20; "local combustion sources" and "industrial emissions" need to be described in more detailed. What kind of combustion sources, just traffic? What industry there are in that area?

Authors' response: The major industrial activities occurring on Jurong Island (i.e., the large industrial zone in Singapore) have been added in Table S1. Additional locations for some of the industries referred in the table have been added on Figure S1a following the second reviewer's suggestion. The abstract has been modified as follow:

Page 1, lines 21-24: "This work provides evidence that over 90% of rBC was originated from local combustion sources with a major part related to traffic and ~30% of them associated with fresh

secondary organic aerosol (SOA) produced under the influences of shipping and industrial emissions activities (e.g., refineries and petrochemical plants) during daytime."

2. Page 4, line 27; "composition –dependent approach" CE values need to be shown in the paper and the uncertainty of the CE has to be evaluated carefully. Since the RIE for sulfate was not determined, the uncertainty analysis is very important.

Authors' response: The CDCE time series has been added in Figure S2 as shown below. Note that CDCE approach is a well-developed data analysis algorithm for correcting the mass concentrations of NR-PM measured by AMS. To further validate this approach for our dataset, the hourly averaged CDCE-corrected $SO_4^{2-}$ and OM mass concentrations are compared with other co-located measurements of $SO_4^{2-}$ (MARGA) and OM (OC/EC analyser) as shown in Figure S3a and b. The strong agreements between the different measurements were observed as discussed in our responses for the general comments above, suggesting that the CDCE derived for our dataset are reasonably good. The manuscript and SI have been revised accordingly as presented in the responses to the general comments.

[Figure]

Figure S2: (a) Time series of composition-dependent collection efficiency ($CDCE_{Mid}$) determined over the entire campaign. Cumulative frequency of (b) aethalometer-based CDCE and (c) OC/EC-based CDCE, and their respective lognormal fitting determined over the entire campaign (with the fitting parameters corresponding to the following equation: $Y = y_0 + A \exp\left\{-\left[\frac{\ln(x/x_0)}{width}\right]^2\right\}$).

3. Page 5, line 13; I don't understand why only few five metal ions were included in the PMF model. I suggest adding more metal ions.

Authors' response: The presence of other metal ions have been investigated in our data analysis but we have decided to exclude those metal ions with signal below the limit of detection. Please refer detail to our response to general comments above.

4. Page 5, line 13-14; metals ions were included in Hz's into the PMF. How does that affect the results of the PMF?

Authors' response: We thank the reviewer for pointing out this. Metal signals in Hz were used for time series comparisons and NWR plots. While for PMF, metal signals were converted to nitrate equivalent mass concentrations assuming their RIE equal to 1. This approach keeps all the signals of organic, $C_n^+$ and metal fragments within similar order of magnitude. Note that we attempted PMF analysis after applying RIEs on metals – $Na^+$, $Rb^+$, $V^+$, $Ni^+$ from Carbone et al., (2015) and $K^+$ from Drewnick et al., (2006) – on the PMF inputs. However, metal signals would drive PMF solutions because of their strong intensities after such RIE correction. Therefore, the RIE of 1 were applied to all metals and the $K^+$ signals were downweight by a factor 2 for the final PMF solution. To make sure such information clearly delivered in the manuscript, the following sentences have been revised or added in the revised version.

Page 5 lines 2-3: "The signals of these trace metal ions were not calibrated and thus their raw signals were used for investigating their temporal variations."

Page 5 lines 24-27: "Lastly, five metal ions ($K^+$, $Na^+$, $Ni^+$, $V^+$, $Rb^+$) were included into the PMF model (Figure S4g-i) as the majority of their signals were higher than their respective limit of detection (Table S2). The metal ion signals were corrected to nitrate equivalent mass concentrations by assuming their RIE values equal to 1 and the $K^+$ signals were downweight by a factor 2"

5. Page 5, line 30; sulfate increase starting from 10:00 can be explained by the active photochemistry. Could you add a diurnal plot showing solar radiation and sulfate in the same figure?

Authors' response: We have added the diurnal variation of solar radiation to Figure 1c to emphasize the hours of day with intense photochemistry as suggested by the reviewer.

[Figure]

Figure 1: [...] (b) Average chemical compositions of NR-PM and rBC with the contribution of the PMF factors to total OA. (c) Diurnal patterns of rBC, total OA, inorganic species and solar radiations (represented by the yellow bars).

6. Page 5, line 30; NH4 ratio and acidity; the discussion on acidity is somewhat unreliable because the RIE for sulfate was not measured. For example in Fig. S4a, it's hard to understand why all the points are below 1:1 line. To me, this seems to be due to the incorrect RIE for sulfate. Could you comment on that?

Authors' response: Regarding the accuracy of $SO_4^{2-}$ mass concentration, we have discussed in our response to the general comments above and we could conclude that the RIE applied for sulfate are reasonable. Note that the main purpose of showing the ratios between measured and predicted $NH_4^+$ (Figure S6a-c) is to highlight the relative changes in aerosols acidity especially when the high $SO_4^{2-}$ loadings were observed. This relative observation should not be affected by the absolute $SO_4^{2-}$ mass concentrations. Furthermore, previous studies have reported that fine particles encountered in Singapore and Southeast Asia are usually acidic (Budisulistiorini et al., 2018; Pye et al., 2019). Budisulistiorini et al., (2018) recently reported an average $NH_{4,meas}/NH_{4,pred}$ ratio of 0.7 ($\pm$0.3) underlining the acidic nature of aerosols outside of hazy periods. Indeed, ammonia emissions caused by biomass burning results in a neutralization of inorganic particles (Budisulistiorini et al., 2018), while usual ammonia level in Singapore remain relatively low compare to other region of the world (Behera et al., 2013).

7. Page 7, line 5; industrial emission; be more specific what kind of emissions

Authors' response: It is rather difficult for us to pin point the exact types of industrial emissions. However, we know that refineries, petrochemical industries and shipping emissions are some of the key emission sources from the industrial zone as reported in Table S1. The sentence has been revised as following.

Page 7 lines 27-29: "It is important to note that the period with elevated concentrations of sulfate and sulfur-containing fragments were synchronized with the sea breeze from the southwest, from which refineries, petrochemical industries and shipping emissions could be carried to our sampling site (Figure 2 and S6e)."

8. Page 7, line 28; define POA, is it only HOA? Isn't COA also primary?

Authors' response: Both HOA and COA are primary in nature. The focus of Section 3.3.1 is combustion emissions, COA are discussed in Section 3.3.2. To avoid such confusion, a sentence has been added at the beginning of Section 3.3.2 to highlight that COA are also one of the major POA component.

Page 9 line 11: "COA is another major POA component identified in this study."

9. Page 7, line 38; "large combustion sources during nighttime" needs to be elaborated since it has such a large effect on diurnal cycle of HOA

Authors' response: We agree with the reviewer that the unknown combustion sources can have substantial impact to the diurnal cycle of HOA. However, their influences were occasionally observed within our sampling period as highlighted in our original discussion. It is difficult to identify the exact emission sources but the elevated concentrations were observed at low wind speed condition, suggesting that they were likely due to local emissions from the city. We do believe it can be related to the local industrial emission, such as the flaring emissions from petrochemical industries. The related text have been revised as following to give the reader some ideas about the potential night time combustion sources.

Page 8 lines 24-35: "High concentrations of rBC, $NO_x$ and HOA were observed during the mid-night between May 25th and 28th, indicating the presence of large combustion sources during nighttime that might impact the air quality at our sampling site occasionally (Figure 1a). This observation also reflected from the diurnal plots (Figures 3 and S8) that the mean mass concentrations of HOA, rBC, $NO_X$ and CO during the mid-night were much higher than their corresponding median values. Figure S8a shows that the highest N:C ratio was observed during the morning traffic peak hours but remained relatively low for the rest of the period. Such observation suggests that the emission characteristics of these unknown combustion sources at nighttime could be different to those associated with traffic emissions during the daytime. Although it is difficult to confirm the origin of those nighttime combustion emissions without further evidence, the observed events were coupled to low wind speed condition, suggesting that they were likely due to local emissions from the city. For example, significant emissions due to flaring from petrochemical industries, which largely depends on plant operation, during the relatively stagnant atmospheric condition could lead to elevated concentrations of the combustion-related species."

10. Page 9, lines 11-15; "organo-nitrate and nitrogen-containing fragments increased during the daytime in this field study." Why there is not more discussion on these species if they were detected? What is their contribution to OA or LO-OOA? Add the ratio of N to C (and S:C) to Figure 3.

Authors' response: N:C ratios have been added to Figure 3 for each factor in the revised version. As shown in the figure below (submitted for Kasthuriarachchi et al., (2020)), nitrogen fragments can be originated from different sources. Since we don't have strong evidence to develop conclusive arguments regarding the nitrogen compounds chemistry from primary emissions and secondary aerosol formation, we have decided to added/revised a few sentences throughout the manuscript to briefly highlight how nitrogen fragments may be associated with different emissions. Given that the contribution of nitrogen-containing fragments were only ranged from 0.6 to 1.4 % for each PMF OA factor, adding detail discussion for nitrogen-containing fragments may dilute the focus of this manuscript.

Page 8 lines 28-31: Discussion for HOA - "Figure S8a shows that the highest N:C ratio was observed during the morning traffic peak hours but remained relatively low for the rest of the period. Such observation suggests that the emission characteristics of these unknown combustion sources at nighttime could be different to those associated with traffic emissions during the daytime."

Page 10 line 10-15: Discussion for LO-OOA - "Furthermore, Kasthuriarachchi et al. (2020) reported that the concentrations of organo-nitrate and nitrogen-containing fragments (i.e., $C_xH_yN^+$ and $C_xH_yNO_z^+$) slightly increased during daytime (e.g., Figure S8a) with LO-OOA, contributing to 30% of observed $C_xH_yNO_z^+$ fragments (Figure S10). This suggests that photo-oxidation of VOCs under high-$NO_X$ condition could be a potential pathway toward LO-OOA formation given that the $NO_X$ concentrations reached up to 160 ppb (Figure 1a, mean = 16.5 ppb) during the campaign."

Page 13, lines 30-32: Discussion for MO-OOA - "Note that MO-OOA contributes to the highest fraction of nitrogen-containing organic fragments ($C_xH_yNO_z^+$ ~ 32% and $C_xH_yN^+$ ~ 46%) that can be generated by biomass burning emissions (Mace et al., 2003; Laskin et al., 2009; Desyaterik et al., 2013; Mohr et al., 2013)."

[Figure]

Figure S10: Contributions of each OA factor to the N-containing fragment groups (Kasthuriarachchi et al., (2020)).

11. Page 11, line 5-; Na as a tracer for fossil fuel combustion. Revise this section by taking into account other sources of Na in urban areas (see general comments).

Authors' response: As shown in our responses in the general comments, our intention is not to use $Na^+$ as a fossil fuel combustion tracer. Rather, we understand that $Na^+$ can be originated from various sources in urban. In particular, we pointed out that $Na^+$ was mainly associated with three OA factors (i.e., HOA, O-HOA, and LO-OOA), which reflects the fact that $Na^+$ could be emitted from different local combustion sources. In addition, although we didn't observe BBOA factor in this study, we have mentioned that biomass burning can be a possible urban source of $Na^+$ in general in the revised version as suggested by the reviewer. A few sentences in Section 3.5.1 have been revised or added as shown below:

Page 12 lines 5-7: "Based on the PMF results that includes trace metal ions, sodium ($Na^+$) was mainly associated with HOA, O-HOA and LO-OOA (Figure 5a) that could be due to different types of fossil fuel combustion emissions (e.g., local traffic, shipping, and various industrial activities) as discussed in Section 3.4."

Page 12 lines 20-23: "Biomass burning can be a possible source of $Na^+$ (Hsu et al., 2011) but no major fresh biomass burning emissions were observed in this study. The MO-OOA factor is suspected to be more influenced by aged regional biomass burning emissions (see more discussion in Section 3.5.2) but $Na^+$ was not strongly associated with this factor."

12. Page 11, Biomass burning tracers, I don't understand why the contribution of biomass burning has been investigated so extensively even though it does not show up in PMF and the metals are not very good tracers for it. Consider making Section 3.5.2. more compact.

Authors' response: The $m/z$ 60 (or $C_2H_4O_2^+$ organic fragments) has been widely used as a tracer organic fragment for identifying BBOA in the PMF analysis. Cubison et al. (2011) have generalized observations from worldwide field data that $m/z$ 60 signature become less and less significant in aged BBOA materials. The implication of such observation is that atmospheric aging of BBOA can lead to formation of MO-OOA (or LV-OOA in earlier studies) but their origins cannot be easily identified in

the conventional PMF analysis. Note that MO-OOA usually represents aged SOA materials in many previous field studies and is a common OA factor that cannot provide much mass spectral characteristic for source identification. How we can better understand the origin of MO-OOA is certainly an important research topic for the atmospheric science/chemistry community.

Since biomass burning emissions are commonly observed in the Southeast Asian in general (not within Singapore), we would like to examine if the regional pollutions, such as aged biomass burning and other anthropogenic emissions, may contribute to the observed MO-OOA in this study. This type of investigation is further motivated by our observation that the $f_{C2H4O2+}$ value of MO-OOA is slightly higher than the background value reported in Cubison et al. (2011) and its moderate correlation with $K^+$ and $Rb^+$. Although the metal ions that we used in this study are not unique for any specific sources, they are more persistent to atmospheric aging compared to their co-emitted OA materials, and thus can provide insight into the potential sources and/or aging history of MO-OOA. As pointed out by the reviewer in the general comment, we have revised the discussion to highlight the fact that $K^+$ and $Rb^+$ can be emitted by both coal combustion and biomass burning. However, it won't substantially change our conclusion that trace metal may allow us to better understand the potential origins of MO-OOA. We do believe the similar approach can also be applied for other SP-AMS measurements in the future.

Since our interested is not limited to the potential influences of aged regional BBOA, we have changed the sub-heading of Section of 3.5.2 to "Potential origins of MO-OOA" in order avoid confusion. Furthermore, the majority of Section 3.5.2 and Section 4 have been rewritten as following,

Section 3.5.2: Page 13, lines 16-37: "Figures 6c and S14b show that high $Rb^+$ and $K^+$ signals were associated with more oxygenated ($f_{44}$ > 0.7) fraction of OA, and moderate correlations between MO-OOA and the two metal were observed ($Rb^+$, r = 0.58 and $K^+$, r = 0.71, Figure S13b). Furthermore, the results of PMF analysis demonstrated that both $K^+$ and $Rb^+$ were mainly associated with MO-OOA (58-66%, Figure 5a) followed by the two combustion-related components (HOA and O-HOA). Although potassium and rubidium are not unique tracers for a specific combustion source, previous studies have shown that these two metals can be largely associated with biomass burning emissions (Artaxo et al., 1993; Lee et al., 2016; Achad et al., 2018). Note that rubidium has been used as a coal combustion tracer in previous studies (Fine et al., 2004; Irei et al., 2014). Unlike ambient OA component, the chemical identities of $K^+$ and $Rb^+$ are unlikely modified by the oxidative aging of aerosol particles. Therefore, a strong temporal correlation between $Rb^+$ and $K^+$ (r = 0.85, Figure 6b) further suggests that they were likely of similar origins in this study.

The regional origin of $K^+$, $Rb^+$, $C_2H_4O_2^+$ and MO-OOA were investigated through their PSCF. Their PSCF graphs (Figures 6d and S15a- c) show several common origins with high probability that the highest concentrations could be influenced by biomass burning events from Indonesia (Figure S12a). Nevertheless, coal-fired power plants are located nearby the identified hotspots of $Rb^+$ and $K^+$ (Figure S12b) so that a regional transport of coal-fired power plant emissions alongside biomass burning plumes were possible. Note that MO-OOA contributes to the highest fraction of nitrogen-containing organic fragments ($C_xH_yNO_z^+$ ~ 32% and $C_xH_yN^+$ ~ 46%) that can be generated by biomass burning emissions (Mace et al., 2003; Laskin et al., 2009; Desyaterik et al., 2013; Mohr et al., 2013). It is important to point out that most of the previous studies usually describe MO-OOA (or LV-OOA in some earlier studies) as aged SOA component without providing further detail on their potential origin and emission characteristics. Our observations underline the possibility of better understanding the origin of the MO-OOA component through measurements of refractory metals even when atmospheric oxidative processing has made the mass spectral features of aged OA materials less distinguishable."

Section 4: Page 14, Lines 24-36: "One of the major challenges in interpreting the PMF results from AMS measurements is to identify the origin and aging history of ambient particles associated with highly oxidized OA components (e.g., MO-OOA in this work), as the mass spectral characteristics of OA converge along with their degree of oxidative aging. In general, the relative intensities of some highly oxygenated fragments (e.g. $CO^+$ and $CO_2^+$) increase continuously while other mass spectral

features (e.g. alkane/alkene patterns from combustion sources) being diminished during the aging processes. Ng et al., (2010) have visualized such phenomena for SOA components on the $f_{43}$-$f_{44}$ space. Furthermore, the *m/z* 60 (or $C_2H_4O_2^+$ organic fragments) has been widely used as a tracer ions for BBOA. Cubison et al. (2011) have generalized observations from worldwide field data that *m/z* 60 signature become less and less significant in aged BBOA materials. In this study, we proposed that the MO-OOA component represented aged OA materials impacted by the regional biomass burning and perhaps coal combustion emissions as MO-OOA was associated with refractory $K^+$ and $Rb^+$. Given the fact that MO-OOA was the major OA components of the total OA (~32%), this result highlights the fact that the regional pollution can affect the air quality in Singapore even though fresh regional biomass burning episodes were not observed during the sampling period."

13. Page 12, line 24-26; "Kasthuriarachchi et al., (2019) also found that MO-OOA was the major contributor to the observed nitrogen-containing organic fragments that could be largely generated by biomass burning emissions" is contradictory to page 9: "Kasthuriarachchi et al. (2019) reported that the concentrations of organo-nitrate and nitrogen-containing fragments (i.e., CxHyN+ and CxHyNOz+) increased during the daytime in this field study, suggesting that photo oxidation of VOCs under high-NO$_X$ condition could be another potential pathway toward LO-OOA formation". Please revise the sentence/sentences.

Authors' response: The elevated concentrations/signals of LO-OOA factor, $C_xH_yN^+$ and, $C_xH_yNO_z^+$ were observed during the daytime, suggesting that the formation of LO-OOA could result from VOCs photo-oxidation under-high $NO_x$ condition. As shown in the response of comment #10, approximately 13-30% of nitrogen-containing fragments were associated with the LO-OOA factor, whereas MO-OOA accounts for ~50% of nitrogen-containing fragments. The first statement is focusing on the diurnal dynamic and the second statement is about the major contributors of those fragments so that they are not contradict to each other. Note that there were no distinct diurnal pattern for MO-OOA factor. The related sentences were slightly modified as following:

Page 10 line 10-15: Discussion for LO-OOA - "Furthermore, Kasthuriarachchi et al. (2020) reported that the concentrations of organo-nitrate and nitrogen-containing fragments (i.e., $C_xH_yN^+$ and $C_xH_yNO_z^+$) slightly increased during daytime (e.g., Figure S8a) with LO-OOA, contributing to 30% of observed $C_xH_yNO_z^+$ fragments (Figure S10). This suggests that photo-oxidation of VOCs under high-$NO_X$ condition could be a potential pathway toward LO-OOA formation given that the $NO_X$ concentrations reached up to 160 ppb (Figure 1a, mean = 16.5 ppb) during the campaign."

Page 13, lines 30-32: Discussion for MO-OOA - "Note that MO-OOA contributes to the highest fraction of nitrogen-containing organic fragments ($C_xH_yNO_z^+$ ~ 32% and $C_xH_yN^+$ ~ 46%) that can be generated by biomass burning emissions (Mace et al., 2003; Laskin et al., 2009; Desyaterik et al., 2013; Mohr et al., 2013)."

14. Page 12, line 31-33 and Fig. S11, Fig. S11 is confusing. It is very difficult to see which trajectories really pass the two fires shown by the MODIS. Could you just select the trajectory for the peak concentrations of m/z 60? Where are the large urban areas in the map? Do the trajectories pass large cities as well? Is it possible that biomass burning emissions were mixed with urban emissions (that increased Rb and maybe also K)? Please add cities to the map. Why Fig. S11(b) and Fig S11(c) are not on the same scale? Now it is difficult to compare (b) and (c). Add lines between markers to m/z 60 and Rb in Fig. S11a.

Author's response: We agree with the reviewer's comment and have decided to remove this figure in the revised version. Instead, we have added a map of all active fires that occurred over the region during the entire campaign (Figure S12a) with the name of large urban areas. As discussed in our above responses for general comments, we are trying to investigate the potential origins of MO-OOA due to the regional emissions based on its moderate correlation with $K^+$, $Rb^+$ and $C_2H_4O_2^+$. Since we

understand that there should be multiple sources of MO-OOA that could be produced from aging of different types of primary and secondary emissions. In the revised manuscript, we would like to highlight the possibilities that regional biomass burning and coal combustion emissions could be linked to the MO-OOA formation. The following text has been added in the manuscript.

Page 13, lines 26-30: "The regional origin of $K^+$, $Rb^+$, $C_2H_4O_2^+$ and MO-OOA were investigated through their PSCF. Their PSCF graphs (Figures 6d and S15a- c) show several common origins with high probability that the highest concentrations could be influenced by biomass burning events from Indonesia (Figure S12a). Nevertheless, coal-fired power plants are located nearby the identified hotspots of $Rb^+$ and $K^+$ (Figure S12b) so that a regional transport of coal-fired power plant emissions alongside biomass burning plumes were possible."

15. Page 13, lines 21-23; "Furthermore, $Na^+$ measured by the SP-AMS can potentially be useful for separating rBC from traffic and biomass burning combustion emissions, which required further investigation." I suggest revising this sentence as the evidences provided by this study are not convincing enough. I think that there are multiple sources for Na in urban areas that should be taken into account.

Author's response: As discussed above, we have revised our discussion regarding $Na^+$. This sentence has been removed in the revised version.

16. Page 13, lines 36-37; "This observation suggest that, in the region, both K+ and Rb+ might be more appropriate tracers for the identification of aged biomass burning than m/z 60." I suggest revising this sentence as Rb may also come from other sources.

Author's response: As discussed above, we have revised our discussion regarding $K^+$ and $Rb^+$. This sentence has been removed from the revised conclusion.

**Technical corrections:**

1. Page 7, line 13; less-oxidized oxygenated OA (LO-OOA)

Author's response: The sentence has been revised accordingly.

2. Supplemental material, could you add a figure showing the contribution of metals to each PMF factor? It would be interesting to see how metals divide between the factors.

Author's response: The figure showing the contribution of metals to each PMF factor is presented in Figure 5a. (Figure 5d in the original version of manuscript).

Author's Response: We conducted four calibrations using ammonium nitrate and Regal black particles as described in the original manuscript. To ensure the quality of calibration, the inter-instrument comparisons have be conducted to show that our SP-AMS calibrations for both rBC and NR-PM are reasonable. The SP-AMS measurements were compared with other co-located measurements from an aethalometer (model AE33, Magee scientific), an OC/EC analyser (Sunset Laboratory) and a Monitor for AeRosols and GAses in ambient air analyzer (MARGA, Metrohm) over the sampling period.

Sulfate ($SO_4^{2-}$) and organic mass concentrations from the SP-AMS (CDCE-corrected) were compared to $SO_4^{2-}$ and OM measured by the MARGA and the OC/EC analyser, respectively (Figure S3a and b in the revised SI). Aside from the good correlations that underline a good temporal response from all instruments, the slopes are in the range of 0.81-0.88, suggesting that the mass concentrations of $SO_4^{2-}$ and organic measured by the SP-AMS are ~12-19% lower than those measured by the MARGA and the OC/EC analyser, respectively. This could be partially explained by the 1 µm cut-size of the SP-AMS, while the two other instruments were measuring $PM_{2.5}$.

The rBC mass concentrations (CE = 0.6) were compared with BC and EC measured by the aethalometer and the OC-EC analyser, respectively (Figure S3c and d in the revised SI). Scatter plots of those comparisons show Pearson coefficients > 0.84, and respective slopes of 0.83 and 1.10, supporting that CE = 0.6 for rBC is a reasonable CE correction approach. The following sentences and Figures S3 have been added to the revised manuscript and SI based on the above observations:

Page 2 lines 7-9: "Other co-located instruments for $PM_{2.5}$ characterizations include an aethalometer (AE33, Magee Scientific), a Monitor for Aerosols and Gases (MARGA, Metrohm) and a semi-continuous organic and elemental carbon (OC/EC) analyser (Sunset Laboratory)."

Page 4 lines 27-29: "The campaign averages of $RIE_{rBC}$ and $RIE_{NH4}$ were 0.15 (±0.04) and 4.24 (±0.04), respectively. The default RIE values of nitrate (1.1), sulfate (1.2) and organics (1.4) were used for respective mass concentration quantification (Jimenez, 2003)."

Page 4 lines 33-38: "The SP-AMS measurements were compared with the sulfate and OM concentrations measured by the MARGA and the OC/EC analyser, respectively, showing strong temporal (r = 0.77 and 0.93) and quantitative (slopes = 0.81 and 0.88) agreements between these measurements (Figure S3a and b). A collection efficiency (CE) of 0.6 was used for rBC quantification

due to incomplete overlap between the laser vaporizer and the particle beam (Willis et al., 2014). The corrected rBC concentrations were also comparable to those measured by the aethalometer (r = 0.96, slope = 0.83) and the EC measured by the OC/EC analyser (r = 0.84, slope = 1.1) (Figure S3c and d)."

[Figure]

Figure S3: Scatter plots of (a) the hourly-averaged $SO_4^{2-}$ mass concentration measured by the SP-AMS and MARGA, (b) and the hourly-averaged OA and OM (estimated from OC concentration using a 2.2 conversion factor) mass concentrations measured by the SP-AMS and OC/EC analyser, respectively. Comparisons of 10-min averaged rBC mass concentrations measured by the SP-AMS with (c) BC and (d) EC mass concentrations measured by the aethalometer and OC/EC analyser, respectively.

We agree with the reviewer that having the knowledge of coating and morphology of the BC-containing particles would be useful in determining the time-dependent CDCE of rBC measurements. However, the SP-AMS configuration used in this study (i.e., dual vaporization scheme) cannot provide further information on coating thickness and rBC morphology. Note that there is no standard approach for determining CDCE of rBC measured by the SP-AMS within the scientific community. In this work, we determined the CDCE of rBC by comparing BC (denoted as $BC_{AE33}$ here after) and EC measured by the aethalometer and the OC/EC analyser, respectively. Comparisons between mass concentrations of our CE-corrected rBC ($rBC_{CE=0.6}$) with $BC_{AE33}$ and EC give good Pearson coefficients of 0.96 and 0.71

and slopes of 0.83 and 1.10, respectively (Figure S3c and d). The histogram of CDCE determined for rBC using the two co-located instruments are reported in Figure S2b and c. The logarithmic distributions are centred on ~0.52 (±0.18) for aethalometer-based CDCE ($CDCE_{AE33}$) and 0.69 (±0.34) for OC/EC-based CDCE ($CDCE_{ECOC}$). These results are in a good agreement with the average CE value of 0.6 suggested by Willis et al., (2014) based on laboratory investigation.

[Figure]

Figure S2: (a) Time series of composition-dependent collection efficiency ($CDCE_{Mid}$) determined over the entire campaign. Cumulative frequency of (b) aethalometer-based CDCE and (c) OC/EC-based CDCE, and their respective lognormal fitting determined over the entire campaign (with the fitting parameters corresponding to the following equation: $Y = y_0 + A \exp\left\{-\left[\frac{\ln(x/x_0)}{width}\right]^2\right\}$).

To evaluate the potential impact of CDCE of rBC and OA on our PMF results, three additional PMF analysis were performed by applying CDCE for rBC and/or OA, while other PMF setting remains unchanged. The metal ion signals were corrected to nitrate equivalent mass concentrations by assuming their RIE values equal to 1.

- Laser-off OA corrected by CDCE that were calculated based on the approach described by Middlebrook et al., (2012) ($CDCE_{Mid}$)

- Laser-on $CDCE_{Mid}$-corrected OA and $CDCE_{AE33}$-corrected $C_n^+$
- Laser-on $CDCE_{Mid}$-corrected OA and $CDCE_{AE33}$-corrected $C_n^+$ and metal ions

Comparisons between the CDCE-corrected results and the corresponding base cases reported in the main text (i.e., laser-off OA, laser-on OA + $C_n^+$ and laser-on OA + $C_n^+$ + metals) are shown in Table S4. Note that all PMF runs lead to five-factor solution (i.e., HOA, O-HOA, COA, LO-OOA and MO-OOA). Below is the brief summary of some major changes:

- OA fragments as PMF input: The time series of OA factors determined with $CDCE_{Mid}$ applied are similar to those without correction (i.e., Pearson coefficients > 0.91). Applying $CDCE_{Mid}$ correction result in 19-27% changes in the mass concentrations of each factor.

- OA, rBC and metals as PMF input: These modifications of PMF input do not make significant impacts on the relative contribution of each factor to the total OA and rBC. Except for the HOA factor, its contribution to the total rBC mass decreased from 44% to 33%. The $C_1^+/C_3^+$ ratios for MO-OOA are much more sensitive to CDCE corrections (i.e., increased from. 0.29-0.54 to 1.57-1.67) compared to other PMF factors. Even without applying CDCE correction, the $C_1^+/C_3^+$ ratios for MO-OOA varied between 0.29 and 0.54. Due to such large variations between cases, in addition to COA, no $C_1^+/C_3^+$ ratios were reported for MO-OOA in the main text.

- The CDCE corrections can affect the contributions of the five metals from each PMF factor to their total signals as shown in Table 1. The changes in the contributions of sodium and nickel from the LO-OOA and HOA factors to their total signals are relatively large compared to other metals and OA factors. It is important to emphasise that a few key observations remain unchanged: (1) $K^+$ and $Rb^+$ are strongly associated with MO-OOA, (2) $V^+$ is mainly associated with LO-OOA, and (3) $Na^+$ is associated with a few OA factors that are related to combustion emissions (i.e., LO-OOA, O-HOA, and HOA).

Overall, applying CDCE corrections for OA, rBC and metals do not result in substantial changes in our interpretations for most of the key observations. The discussion and conclusion developed based on the distribution of metals to different OA factors remains unchanged. Therefore, the original PMF results are used as a base case in our discussion in the main text. Table 1 shows the possible ranges of different parameters based on the results obtained from the CDCE-corrected PMF analysis. Table S4 summarizes the results of pre- and post-CDCE-corrected PMF analysis (Pre-CDCE: PMF solution obtained by the CDCE-corrected input matrices, Post-CDCE: PMF solution corrected by CDCE).

The above information has been added to the supplementary information. The following sentences have been added in the revised experimental section so that the readers can follow the detail of PMF analysis and results.

Page 5, lines 34-37: "The time dependent collection efficiency (or CDCE) of rBC were determined based on the BC measured by the aethalometer (Figure S2b). Additional PMF analysis, with CDCE applied on the input matrix for both OA and refractory components, were conducted (see details in Text S1 of the supplementary information). The PMF results were compared with those obtained without CDCE applied on the input matrix as shown in Table 1 and S4."

Table 1: Top: Carbon fragment ratios observed by the laser-on measurement for each PMF factor. The underlined number represents the results with both rBC fragments and trace metal ions included as PMF input. Bottom: Contribution of each PMF factor to the total signal of specific metal ions. For the entire table, the values in parenthesis are the one obtained from PMF with CDCE-corrected input matrix (see details in Text S1 and Table S4 of the supplementary information).

| PMF factors | HOA | O-HOA | COA | LO-OOA | MO-OOA |
|---|---|---|---|---|---|
| Carbon fragments ratios | | | | | |
| $C_1^+/C_3^+$ | 0.66 (0.63)
 0.65 (0.62) | 1.00 (0.90)
 1.00 (0.89) | NA* | 0.81 (0.88)
 0.79 (0.85) | NA# |
| $C_2^+/C_3^+$ | 0.38 (0.39)
 0.38 (0.39) | 0.41 (0.40)
 0.41 (0.40) | NA* | 0.39 (0.41)
 0.41 (0.40) | NA# |
| Contribution of each factor to the total signal of specific metal ions (fraction) | | | | | |
| $Na^+$ | 0.35 (0.22) | 0.14 (0.17) | < 0.01 (< 0.01) | 0.45 (0.58) | 0.06 (0.03) |
| $K^+$ | 0.23 (0.23) | 0.19 (0.18) | < 0.01 (< 0.01) | <0.01 (0.05) | 0.58 (0.54) |
| $V^+$ | 0.21 (0.08) | 0.09 (0.16) | < 0.01 (< 0.01) | 0.70 (0.76) | < 0.01 (< 0.01) |
| $Ni^+$ | 0.38 (0.22) | 0.20 (0.22) | < 0.01 (< 0.01) | 0.29 (0.45) | 0.13 (0.11) |
| $Rb^+$ | 0.15 (0.15) | 0.19 (0.19) | < 0.01 (< 0.01) | <0.01 (0.01) | 0.66 (0.65) |

* None of the refractory $C_n^+$ fragments were associated with the COA factor
**Large variations of $C_n^+$ fragment ratios between cases, hence the ratios were not reported for MO-OOA**

Table S4: Comparisons of the PMF results obtained from different CDCE correction approach (Pre-CDCE: PMF solution obtained by the CDCE-corrected input matrices, Post-CDCE: PMF solution corrected by CDCE). The post-CDCE solutions represent the corresponding base cases reported in the main text, Figure 3, and Figure S4.

| PMF factors | LO-OOA | | | MO-OOA | | | COA | | | O-HOA | | | HOA | | |
|---|---|---|---|---|---|---|---|---|---|---|---|---|---|---|---|
| **Laser status** | OFF | ON | | OFF | ON | | OFF | ON | | OFF | ON | | OFF | ON | |
| **Type of PMF** | OA | $+ C_n^+$ | $+ C_n^+ +$ metal | OA | $+ C_n^+$ | $+ C_n^+ +$ metal | OA | $+ C_n^+$ | $+ C_n^+ +$ metal | OA | $+ C_n^+$ | $+ C_n^+ +$ metal | OA | $+ C_n^+$ | $+ C_n^+ +$ metal |
| **Contribution to total OA mass (%)** | | | | | | | | | | | | | | | |
| **Post-CDCE** | 10.4 | 12.6 | 13.1 | 32.1 | 24.5 | 26.7 | 11.7 | 14.6 | 15.4 | 26.4 | 22.2 | 22.0 | 19.4 | 23.7 | 22.8 |
| **Pre-CDCE** | 12.4 | 12.3 | 12.1 | 30.8 | 28.5 | 29.6 | 12.6 | 16.3 | 17.8 | 21.0 | 20.6 | 19.7 | 23.2 | 22.2 | 21.4 |
| **Contribution to total rBC mass (%)** | | | | | | | | | | | | | | | |
| **Post-CDCE** | NA | 29.2 | 30.1 | NA | 6.2 | 6.4 | NA | 1.3 | 1.4 | NA | 20.1 | 20.7 | NA | 43.1 | 44.4 |
| **Pre-CDCE** | NA | 29.9 | 31.5 | NA | 7.4 | 8.8 | NA | 1.8 | 2.3 | NA | 26.5 | 24.4 | NA | 34.4 | 33.0 |
| **$C_1^+/C_3^+$** | | | | | | | | | | | | | | | |
| **Post-CDCE** | NA | 0.81 | 0.79 | NA | 0.54 | 0.29 | NA | NA | NA | NA | 1.00 | 1.00 | NA | 0.66 | 0.65 |
| **Pre-CDCE** | NA | 0.88 | 0.85 | NA | 1.76 | 1.57 | NA | NA | NA | NA | 0.90 | 0.89 | NA | 0.63 | 0.62 |
| **Time series - correlation coefficient (r)** | | | | | | | | | | | | | | | |
| **Post- vs. Pre-CDCE** | 0.98 | 0.97 | 0.91 | 0.99 | 0.99 | 0.99 | 0.98 | 0.97 | 0.96 | 0.89 | 0.93 | 0.94 | 0.99 | 0.97 | 0.97 |
| **Time series – slope** | | | | | | | | | | | | | | | |
| **Post- vs. Pre-CDCE** | 1.17 | 1.06 | 1.06 | 0.96 | 0.99 | 1.09 | 1.07 | 0.86 | 0.83 | 0.81 | 1.05 | 1.11 | 1.27 | 1.04 | 1.10 |
| **Normalized mass spectra - correlation coefficient (r)** | | | | | | | | | | | | | | | |
| **Post- vs. Pre-CDCE** | 1.00 | 0.95 | 0.94 | 1.00 | 1.00 | 0.99 | 1.00 | 0.93 | 0.92 | 1.00 | 0.99 | 0.99 | 0.97 | 0.99 | 0.99 |

2) In Sect. 3.4, the C1/C3 analysis should include a discussion about how uncertainty in the contribution of C1+ from non-refractory organic aerosol affects the interpretation of the ratio, particularly since a constant C1/C3 ratio was used to as a correction factor (Sect. 2.2). It may be beneficial to investigate the ratio of other Cx ions (e.g., C4/C3) to provide insight into the possible contribution of organics to C1.

Author's response: The $C_1^+/C_3^+$ ratio of 0.625 is used to calculate the total rBC mass concentrations. However, this correction was not applied when generating the PMF matrix. Instead, we applied Wang et al (2018) method to obtain the refractory fraction of $C_n^+$ for individual OA factor (i.e., correcting total $C_n^+$ fragments obtained from laser-on PMF by subtracting the organic fraction of $C_n^+$ ions retrieved from laser-off PMF). The lower intensities of the $C_4^+$-$C_9^+$, relatively to $C_1^+$-$C_3^+$ ions, can be strongly affected by neighbouring peaks (e.g. $C_4^+$ and $SO^+$), hence we did not report their values in the manuscript. To make this information clear, the related sentence has been revised as following.

Page 5 line 21-24: "Note that the interference of non-refractory organic signals on refractory $C_n^+$ fragments were subtracted based on the method described in Wang et al. (2018), and hence the $C_1^+$ fragment was not corrected based on the $C_3^+$ fragment, in the PMF analysis."

3) Sect. 3.5.1 would be strengthened by a more detailed discussion regarding urban sources of metals and including more metals in the analysis.

Author's response: Same response as reviewer #1, it is important to include all the detectable metal signals in the PMF analysis if possible. However, we only reported metals with their average values higher than the limit of detection (Table S2) determined by the particle-free air to ensure the quality of our data analysis. Nickel is a marginal case but a large fraction of its signal remain above its detection limit for investigating the V+/Ni+ ratio. To make this point clear, we have revised the sentence as shown below.

Page 5 lines 24-25: "Lastly, five metal ions ($K^+$, $Na^+$, $Ni^+$, $V^+$, $Rb^+$) were included into the PMF model (Figure S4g-i) as the majority of their signals were higher than their respective limit of detection (Table S2)."

Furthermore, we have followed the recommendations from reviewer #1 to include more possible sources of metals, such as biomass burning emissions for $Na^+$ and coal combustion emissions for $Rb^+$, in our discussion. Below are a few sentences/paragraphs that we have modified. Please see more detail of modification in the revised version highlighted in the point-by-point responses to reviewer #1.

Page 12 lines 5-7: "Based on the PMF results that includes trace metal ions, sodium ($Na^+$) was mainly associated with HOA, O-HOA and LO-OOA (Figure 5a) that could be due to different types of fossil fuel combustion emissions (e.g., local traffic, shipping, and various industrial activities) as discussed in Section 3.4."

Page 12 lines 20-23: "Biomass burning can be a possible source of $Na^+$ (Hsu et al., 2011) but no major fresh biomass burning emissions were observed in this study. The MO-OOA factor is suspected to be more influenced by aged regional biomass burning emissions (see more discussion in Section 3.5.2) but $Na^+$ was not strongly associated with this factor."

Page 13, lines 27-30: "The regional origin of $K^+$, $Rb^+$, $C_2H_4O_2^+$ and MO-OOA were investigated through their PSCF. Their PSCF graphs (Figures 6d and S15a- c) show several common origins with high probability that the highest concentrations could be influenced by biomass burning events from Indonesia (Figure S12a). Nevertheless, coal-fired power plants are located nearby the identified

hotspots of Rb[+] and K[+] (Figure S12b) so that a regional transport of coal-fired power plant emissions alongside biomass burning plumes were possible".

Additionally, the potential contribution of oceanic sources to Na[+] should be discussed in more detail. While the authors state that Na+ and Cl- exhibit poor temporal correlation, I assume this is for the whole data set. If one filters by wind direction (and thus sea breeze) does this still hold? Additionally, is there an interpretation for the large contribution of Na+ in LO-OOA? Finally, does the fact that different C1/C3 ratios were seen for LO-OOA and HOA (the factors with the strongest Na+ contribution) tell us anything about the different sources of Na+?

Author's response: Thanks for the suggestion on the data analysis. Firstly, the comparison between Na[+] and Cl[-] under sea breeze conditions do not improve the correlation, with r < 0.1 (n=3384). The large contribution of Na[+] in LO-OOA can be due to the fact that this OA factor is associated with industrial emissions. In particular, LO-OOA had the second largest contribution to the total rBC, suggesting that at least a fraction of rBC from industrial emissions could act as effective condensation sinks of LO-OOA produced via the photochemistry along their dispersion as discussed in Section 3.4 of the original manuscript. Finally, the different $C_1^+/C_3^+$ ratios between LO-OOA and HOA can provide insight to the possible sources of Na[+] as discussed in Sections 3.4 and 3.51. To better connect the two Sections, a sentence has been added in Section 3.5.1, and the paragraph in Section 3.4 has been revised as following.

Section 3.5.1, page 12, lines 5-7: "Based on the PMF results that includes trace metal ions, sodium (Na[+]) was mainly associated with HOA, O-HOA and LO-OOA that could be due to different types of fossil fuel combustion emissions (e.g., local traffic, shipping, and various industrial activities) as discussed in Section 3.4."

Section 3.5.1, page 12, Lines 18-20: "However, it is important to emphasize that Na[+] and Cl[-] exhibit rather poor temporal correlations (r < 0.30) for both laser-off and laser-on data regardless the influences of sea breeze."

Section 3.4, page 11, lines 13-27: "The $C_1^+/C_3^+$ ratios of rBC associated with HOA was 0.66 (±0.07) , which was within the range of those emitted from aircraft-turbine, regal black (i.e., a BC standard for SP-AMS calibration) and particles produced by a propane diffusion flame (0.50 – 0.78) and, more importantly, close to those reported for diesel engine exhaust (Carbone et al., 2019; Corbin et al., 2014; Onasch et al., 2012). Furthermore, the size distribution of unit mass resolution data shows lower $m/z$ 12-to-$m/z$ 36 ratios (a proxy for $C_1^+/C_3^+$) for particles with $d_{va}$ smaller than 100 nm (Figure S11a). This suggests that rBC particles with relatively small diameter were mainly associated with fresh traffic emissions (Massoli et al., 2012). Our results illustrate that rBC transported from industrial area and shipping ports gave $C_1^+/C_3^+$ ratios closer to unity (i.e., LO-OOA = 0.79 (±0.10) and O-HOA = 1.00 (±0.11), which is similar to the previous observations for soot particles emitted from a marine engine using heavy-fuel-oil (Corbin et al., 2018), rBC-containing particles emitted from chemical and petrochemical industries (Wang et al., 2018), and rBC with high fullerene content (Canagaratna et al., 2015a; Corbin et al., 2014). The NWR and diurnal plots of $C_1^+/C_3^+$ ratio (Figure 4c and d) clearly show that rBC with higher $C_1^+/C_3^+$ ratios were transported to the site by sea breeze from the southwest direction, which are consistent with our PMF results that O-HOA and LO-OOA were influenced by industrial emissions and were associated with rBC with higher $C_1^+/C_3^+$ ratios compared to other OA components."

4) To me, it seems that the last sentence of the introduction and the last paragraph of the conclusions overstate the implications and applications of the manuscript. Optical properties of aerosols were not

discussed in the manuscript, so the inclusion in the introduction and conclusion seems out of place. Moreover, the work of Kasthuriarachchi et al (2019) is under review so the inclusion here is somewhat misleading. I suggest that the authors consider revising this paragraph and the last sentence of the introduction to better reflect the material covered in this study.

Author's response: The last sentence of the introduction has been removed as suggested. In addition, the last paragraph of conclusion has been revised as following. In particular, the observations from Kasthuriarachchi et al (2020) has been removed from the conclusion.

Page 14, lines 37-38 to Page 15, lines 1-9: "More broadly, the improved source identification for OA and rBC can provide useful information to further investigate the effects of atmospheric aging on their physio-chemical properties. For example, this work highlights the potential influences of regional biomass burning and coal combustion emissions to MO-OOA component, which may provide important insight into how light absorbing properties of OA (i.e., brown carbon) evolve with transport and aging. Recently, Dasari et al. (2019) provided field evidence for the bleaching of brown carbon during their transport by photo-oxidation and that photo-dissociation can occur in the South Asian outflow based on measurements near and away from specific combustion sources, including biomass burning and traffic. Furthermore, this type of PMF analysis could be applied to analyze the sources and characteristics of rBC-containing particles exclusively (i.e., rBC core with organic coatings) in order to advance our understanding on the effects of primary emissions and/or atmospheric processing on BC light absorption enhancement caused by the lensing effect."

**Minor comments**

-It would be helpful to explicitly pinpoint some of the main industrial sources (such as the oil refinery discussed in the introduction and shipping ports) in Figure S1a.

Authors' response: The major industrial activities occurring on Jurong Island (i.e., an industrial zone located at the southwest direction of sampling site) have been added in Table S1. Additional pins have been added to illustrate their locations in Figure S1a.

-Page 7 lines 12-15: Regarding the production of organo-sulfur compounds in acidic sulfate plumes – this seems rather poorly supported at the moment. As it does not advance the main focus of the paper, I suggest removing. -Figure S4a the legend (colours based on CH2SO) does not match the caption (data originating from SW and other directions).

Authors' response: We agree with the reviewer and have removed the sentence regarding the production of organo-sulfur compounds in acidic sulfate plumes. The caption has been corrected to match the figure (Figure S6 in the revised version).

-Figure S4d needs a color scale. The 2 modes (discussed at the end of page 6) are also not apparent from this figure, perhaps due to the color scale used. Currently it looks to me more like the distribution broadens when the concentration increases in the middle of the day.

Authors' response: The colour scale has been added to Figure S6a (former Figure S4d). We agree with the reviewer that the distribution of sulfate particles broaden in the middle of the day. It is worth noting that the Figure S6a is the average of the entire campaign. If we separate the strong sulfate plumes from

the entire dataset, Figure S6d (former Figure S4b) shows that the sulfate particles size is smaller for the strong sulfate plumes compared to the rest of periods. The sentences have been revised as following.

Page 7, lines 21-26: "This particle acidity dependence seems to coincide with diurnal variations of $SO_4^{2-}$ size distribution. Throughout the day, relatively large sulfate particles are observed with their vacuum aerodynamic diameter ($d_{va}$) peaked at ~400 nm, whereas those encountered during the sea breeze present a broader mode with $d_{va}$ ranging between 200 and 400 nm (Figure S6a and d). This suggests that a large fraction of acidic $SO_4^{2-}$ particles observed during the sea breeze period were freshly formed in the atmosphere."

-Discuss figures in order (e.g., Fig. S6 is currently referenced before S5). Cite panels of figures in order (for instance, the panels of Fig. S4 are cited out of order).

Authors' response: Both manuscript and SI figures have been re-arranged accordingly.

-Figure S5: The wording in the caption is unclear, specifically "... both filtered from 25th to 28th of May, over the campaign." Is "both" just COA and HOA? Does it mean that only the 25th to 28th was included?

Authors' response: We agree with the reviewer and have reworded the caption as follow:

Figure S8 (former Figure S5): "Figure S8: Diurnal cycles of (a) N:C (laser-off mode), (b) $NO_x$, (c) CO, (d) COA, (e) HOA, and (f) $O_3$ over the campaign. The plain and dotted lines represent the medians and averages values, respectively. The shaded regions represent the 25th and 75th percentiles. Note that data between May 25th to 28th were excluded for HOA and COA due to the strong unknown emission during the nighttime."

-In Sect 3.5.2, why use m/z 60 rather than the HR ion (C2H4O2+)?

Authors' response: The $f_{60}$ was calculated using organic fragment signals at $C_2H_4O_2^+$. We have replaced $f_{60}$ and m/z 60 by $f_{C2H4O2+}$ and m/z $C_2H_4O_2^+$, respectively, in Section 3.5.2 and corresponding figures.

-Page 10 line 27: rather than "inferior" use "smaller than"

Authors' response: The correction has been made.

-Page 12 line 23: I do not find the similar size distributions for m/z 39 and 60 to be convincing evidence for a BBOA source. m/z 60 is the only individual organic ion shown in Fig. S8b. How unique is the size distribution for m/z 60 compared to other individual organic ions?

Authors' response: We agree with the reviewer so that the sentence has been removed. However, we have decided to keep the Figure in SI as literature information.

-Page 12 line 19: should this be Figure 5d rather than 3c and 4b?

Authors' response: We thank the reviewer for pointing out this error. It has been corrected in the revised version.

**References:**

[revised manuscript text omitted]